# The art of valuation: Using visual analysis to price classical paintings by Swedish Masters

**Adri De Ridder**[1,2☯], **Steffen Eriksen**[2☯]*, **Bert Scholtens**[2,3☯]

**1** Department of Business Studies, Uppsala University, Campus Gotland, Visby, Sweden, **2** Department of Economics, Econometrics & Finance, Faculty of Economics and Business, University of Groningen, Groningen, The Netherlands, **3** School of Management, University of Saint Andrews, St. Andrews, Scotland, United Kingdom

☯ These authors contributed equally to this work.
* s.eriksen@rug.nl

## Abstract

This study seeks to address the difficulty of pricing art and the limitations of conventional valuation models by using visual analysis to determine the price of paintings. We examine a large hand-collected sample of classical paintings by Swedish Masters, categorize them based on various dimensions, and reduce measurement error by visually examining and classifying each painting into a theme. We compare this 'visual' approach with the conventional 'terminological' approach. We find that the technique, theme, and auction house all have a substantial impact on the price. We argue that a visual inspection should take precedence over analysis based on the artwork's title. This is because the latter leaves many artworks unclassified and results in a systematic bias. The study demonstrates the importance of using art-informed characteristics to reduce measurement error in pricing paintings.

## 1. Introduction

*"What's in a name? That which we call a rose by any other name would smell just as sweet."* William Shakespeare used this line in his play 'Romeo and Juliet' to convey that the naming of things is irrelevant. However, the moment we start inferring an object's category based on its name, then it is no longer true. In the world of paintings, the name of a painting may not be important for a potential buyer at an auction. For instance, the painting "No. 7" by Mark Rothko would still have sold for 82.5million USD in November 2021 at a Sotheby's auction, had the name been Peach-Lemon-Orange instead. However, when researchers study the drivers of art prices, the name becomes important as it is often used for thematic classification of the painting, thus making the name of a painting relevant, contradicting meaning conveyed by the words of William Shakespeare.

Paintings are economic goods whose values can vary tremendously. Previous literature has devoted a lot of attention to describing the difficulties in pricing paintings. One such difficulty is the shared uncertainty observed in the art market about the attribution of a work of art [1]. Another difficulty lies in identifying the exact drivers of the prices of paintings, and assessing their respective impacts on the price. Prior studies show that the selection of auction house,

**Data Availability Statement:** All relevant data are within the paper and its Supporting Information files.

**Funding:** The authors received no specific funding for this work.

**Competing interests:** The authors have declared that no competing interests exist

artist, auction method, theme, size of the piece, and when the work is sold are variables that influence the auction price (e.g., [2, 3]). However, two caveats from prior studies need to be addressed with respect to the methods employed. First, most prior studies rely on a terminological approach, using the title of the painting to categorize each painting into their respective thematic categories (see [2, 4–7]). These studies usually employ an algorithmic search string for keywords, which then enables thematic classification of each painting. However, the title of the painting can be misleading, be untitled, and can change over time. As such, the thematic classification based on a terminological approach, may not be consistent (and even incorrect) in the first place. Such potential misclassifications can result in measurement error, which would bias any estimate of the price of a given painting (see also [8, 9]). Second, if only sold paintings would be analyzed, there might a sample bias present, as paintings brought to auction but left unsold are excluded. We address these challenges, by visually inspecting each painting in our sample for thematic classification, and by having a sample of both sold and unsold paintings using a sample of sold and unsold paintings. We compare this approach with the terminological analysis as employed in Renneboog and Spaenjers [4].

Our sample consists of 6,566 classical paintings by Swedish Masters that were offered at classical art auctions by the three leading Swedish auction houses. Although no precise definition of a timeline is applicable, most of the paintings can be attributed to the period 1850 to 1920 (see [10]). From our dataset, numerous examples of potential misclassification via the terminological approach exists. For example, Anders Zorn's "Red Sand" shown in figure one (left), and Anders Zorn's "Old Mirror" might as per a terminological approach get a thematic classification of Landscape and Object respectively (or be left as 'unclassified'; when applying the terminological approach as in [4], both paintings are classified as 'unclassified'), but not a nude as they evidently are (see also Appendix A in S1 Appendix for more examples). Therefore, from an econometric perspective, the terminological approach to classifying paintings may result in measurement error (see [11]). To address this, we suggest to also use information that informs about artistic properties of the paintings studied. This aligns with recent approaches in the literature, which increasingly use artificial intelligence (for example, [12–14]. Tan et al. [14] classify fine-art paintings using neutral network analysis. Choi et al. [12] measure by the singular value decomposition (SVD) entropy of the painting image and find that this has a highly significant impact on sales prices. Sheppard [13] aims at measuring the complexity and content of art images. His approach uses information theory in combination with machine learning algorithms to generate measures for image content measurements for more than 300 thousand works sold at auction. Sheppard finds that art buyers have a preference for image complexity and are willing to pay for it [13]. Our approach aims to complement this line of research. We argue that our 'visual' approach is superior to the terminological approach as each painting is visually examined and then classified into a theme with input from art specialists. Our approach also includes untitled paintings, which go unclassified in a terminological analysis. As such, we feel our pricing model is better able to identify the price drivers of paintings.

In general, the hedonic regression model is the econometric technique used to compute and infer characteristics related to price. This is the workhorse model used in almost all previous studies that try to determine what drives the price of art (see, e.g., [15]). Here, the price of a painting is related to a set of continuous variables, primarily binary variables or attributes that reflect each painting's characteristics. We use a Heckman [16] model, which is suitable to correct for endogeneity induced by sample selection bias (see [17]). Studies prior to Collins et al. [17] often rely on samples comprising only sold paintings, whereas our sample includes both sold and unsold paintings. The two-stage Heckman approach to correct for sample-induced endogeneity involves first estimating the probability of a painting entering the sample; the second stage predicts the price.

We consider two important features of this specific market. First, we examine the relation between painting characteristics and hammer price. Second, we examine differences between paintings that have sold and those that have not. Our empirical analysis establishes highly relevant differences between sold and unsold paintings, as well as differences among the auction houses. Further, it supports existing evidence that oil paintings generally sell at a premium compared to paintings suing other media (e.g., [18]). We also find strong support for our conjecture that the theme of a painting relates to the hammer price. In particular, we find that paintings classified as nudes sell at a substantial premium (see also [19]).

Our results make two contributions to the literature on the pricing of paintings: (i) using art-informed thematic classification is superior compared to terminological analysis as it allows the analysis of all paintings, whereas terminological analysis leaves many artworks as 'unclassified', and (ii) we add to the existing evidence showing that a sample selection bias exists. That is, there are significant differences between sold and unsold paintings, which should be accounted for when explaining the price that a painting fetches at an auction.

This paper proceeds as follows. Section 2 describes our model, data and methodology. Section 3 presents our results. Finally, Section 4 sets forth our conclusions.

## 2. Materials and methods

This section motivates the model to examine the relationship between price and painting characteristics, introduces the sample, and discusses the estimation strategy.

### 2.1 Model

To explain the price of a given asset, usually two approaches are considered, the hedonic model and the repeated sales model. Waugh [20] introduced hedonic regressions in examining the factors that influence the price of asparagus. Since then, the hedonic regression methodology has been used in studies relating to a wide range of products, such as automobiles [21], computers [22], arts [23], and real estate [24]. We infer the hedonic price by relating a painting to its attributes [25]; this is the standard procedure used in most prior studies on paintings (see [15]). As each painting is unique, this approach enables us to quantify the effect of each attribute in relation to the hammer price. This method makes efficient use of all available data, and can therefore provide more reliable estimates [4]. Further, sample selection bias can be controlled for provided that information is available for unsold assets. Some issues with the hedonic approach include assumptions that there are no changes in the parameters across the sample period, and that the model follows an implicit functional form [26, 27].

The repeated sales model, commonly applied in the housing market [28, 29], has also been applied to the art market [4, 30, 31]. Unlike the hedonic model, the repeated sales model confines the sample to consider only paintings sold at least twice within the sample period. The main advantage of the repeated sales model is that is does not require information about specific painting attributes. However, this method is prone to sample selection bias, as it is based on only a subsection of the overall sample. That is, paintings that are sold multiple times may have different characteristics than paintings that only sell once or not at all within the given sample period. In addition, it is difficult to control for market trends.

Therefore, our analysis relies on a hedonic pricing approach. The hedonic regression model relies on each painting's characteristics and binary variables, where we estimate the following model:

$$\log(P_{i,t}) = \alpha + \beta_i X_{i,t} + \varepsilon_{i,t}, \tag{1}$$

where dependent variable $P_{i,t}$ is the natural logarithm of the inflation-adjusted hammer price $P$

in Swedish Krona (SEK) of painting $i$ in year $t$; $X_{i,t}$ refers to a vector of specific attributes of each painting $i$ in year $t$; and $\varepsilon_{i,t}$ is the error term. Each of the $\beta_i$ coefficients reflect the additional value a particular value-determining attribute has on the price of a given painting.

A relevant concern in assessing prices of paintings is selection bias. This is due to the fact that not all paintings at an auction are actually sold. Estimating Eq (1) using OLS on a random sample, under the assumptions that the errors, $\varepsilon$, are i.i.d., provides unbiased and efficient estimates of $\beta_i$ (painting characteristics) on the dependent variable of interest (natural logarithm of inflation-adjusted hammer price). However, the sample selection observed in this case involves incidental truncation, as the dependent variable is observed only if it has a value and if one or more other variables take on particular values [32]. That is, the hammer price is available only for paintings sold. Therefore, using a sample consisting only of paintings sold at auction would lead to a non-random sample. Further, paintings sold at auction may differ in their characteristics from paintings that are not sold.

Heckman [16] introduced a two-step method to correct for endogeneity induced by this sample selection problem. The Heckman model is designed to account for endogeneity when it is sample-induced, and not for endogeneity that might arise from other sources. Should there be endogeneity from other sources, then other models such as a 2SLS are preferable to a Heckman model [32, 33]. In the first stage, a probit model is used to estimate the probability of an observation entering the sample with a hammer price. The second stage employs OLS to predict the dependent variable. This two-stage method accounts for potential sample selection bias by using the first-stage probit in combination with the second stage OLS regression to create a selection parameter, known as the inverse Mill's ratio (IMR). This selection parameter is included as a regressor in the second stage OLS regression, and is commonly referred to as lambda ($\lambda$). A number of studies use this method on different samples of paintings. For example, Collins et al. [17, 26] apply a Heckman model on a sample of symbolist paintings, comparing it against a conventional hedonic model. Marinelli and Palomba [18] also apply a Heckman approach to control for sample selection bias for a sample of Italian contemporary art (see also [3, 34, 35] for more applications). In a regression framework, the Heckman two-step method can be expressed by the following set of equations:

$$\log(P_{i,t}) = \alpha \; + \; \beta X_{i,t} \; + \; \lambda_{i,t} \; + \; \varepsilon_{i,t} \tag{2}$$

$$s_{i,t} = \alpha + \beta X_{i,t} + \gamma Z_{i,t} + \omega_{i,t}, \tag{3}$$

where $X$ and $\varepsilon$ in Eq (2) are defined as in Eq (1); $\lambda_{i,t}$ is the IMR, and captures any potential sample selection bias. The dependent variable $s_{i,t}$ in Eq (3) is a binary outcome variable equal to 1 if a painting is sold at a given auction, i.e., entered the sample as expressed in Eq (2), and is zero otherwise. $Z$ is the vector of exclusion restrictions, while $X$ is the same vector of specific attributes of each painting as in Eq (2), and $\omega_{i,t}$ is the error term. The model is estimated via maximum likelihood, which provides estimates of both equations and guarantees consistency and asymptotic normal efficiency [36]. The model expressed via Eqs (2) and (3) can thus be seen as a hedonic regression, where the selection parameter, IMR, corrects for potential sample selection bias.

In estimating a Heckman model, it is important to have at least one exclusion restriction [37]. The exclusion restrictions employed in the model should impact the observation appearing in the sample, but not the main outcome of interest in the second stage OLS regression. We also use one exclusion restriction. It is present in the first step, but not in the second step of the regression analysis. The exclusion restriction should affect the *probability* of a painting being sold, but not the *price* of a painting being sold. Our exclusion restriction is the fraction

of paintings sold at the previous auction. Based on information obtained directly from the auction houses, the fraction of paintings sold at auction has been rather stable over time, and is seen as a good predictor of the fraction of paintings that will sell at the next auction.

The hedonic regression model includes a number of characteristics related to each of the paintings at auction. We consider three characteristics for each painting: artist, painting, and sales characteristics. For artist characteristics, we consider gender and whether the given painting is signed [34, 38, 39]. Furthermore, we also include artist fixed effects to account for artists' reputation and other personal traits [5, 34, 40, 41]. For painting characteristics, we control for theme, technique, and painting size (see e.g., [3, 34]. For sales characteristics, we control for the auction house at which the sale took place, year of sale, and whether it was a winter or summer auction (see also [35, 38]).

In practice, for a selection bias to exist, it is necessary to show that the exclusion restriction has a significant impact on the dependent variable in the first stage. Additionally, there should be a non-zero correlation between the error terms of the two equations (Eqs (2) and (3)). The correlation between the two error terms is known as rho ($\rho$), and IMR is a function of this rho. In other words, we expect a significant IMR. Following Trevis Certo et al. (2016), we also evaluate the correlation between IMR and our exclusion restriction to better understand the strength of the restriction.

## 2.2 Data

Art pieces are unique. We are therefore able to compare pieces of art only if we consider a sample that is as homogeneous as possible. This leads us to focus on classical art by Swedish Masters. That is, we focus on a select group of artists. The sample paintings were offered via the three leading Swedish auction houses (Bukowskis, Stockholms Auktionsverk and Uppsala Auktionskammare). These auction houses are quite selective in the paintings included in their auctions to ensure that they auction only classical art from Swedish Masters. To investigate the determinants of prices of the works of Swedish Masters, we rely on data from auction catalogues and auction house web pages between 2010 and 2020. In general, auctions of Swedish Masters are held bi-annually (June and December) by all three houses. This results in a total of 66 auctions for which we have data. We hand-collect detailed information about each painting from auction catalogues and from the web pages of the auction houses. Although there are online databases with information about art paintings, we refrained from using these. This was motivated by shortcomings in these databases. For example, some paintings may have received revised prices, or there would be data on paintings which was withdrawn prior to an auction, and thus should not be included in the dataset. Further, these databases' thematic classification of paintings are problematic, as they are often done via a terminological approach. An approach, which we argue is inferior to an art-informed thematic classification.

Following advice from art specialists at the University of Uppsala, we categorized each painting into one of four groups, based on the painting technique used as reported by the auction house: (a) oil paintings, (b) water colors, (c) pastels, and (d) mixed. We exclude paintings where the artist is not clearly identified, when there is an unidentified technique, when several different paintings with different artists are combined into one single painting, paintings on textiles, paintings on door shields, etchings, and painted etchings. As shown by Holub et al. [42], such items have their own price dynamic, and often trade in a separate section of the auctions. Considering paintings from the four groups assures the homogeneity of our sample. In addition, our approach follows the literature [7, 38, 39] by monitoring artist characteristics as well as the size and theme of each painting.

Table 1 provides an overview of the final sample after implementing the filters described. This sample consists of 6,566 paintings, of which 4,873 were sold and 1,693 were not sold. We exclude a small number of paintings withdrawn from auctions prior to the event when their provenance was not clear. The first three columns show, for each year, number of paintings offered, number of artists, and number of sold paintings, respectively. Columns 4–7 report number of paintings grouped by technique and the last row in the table gives the total for each group. Our total sample comprises 6,566 paintings by 416 artists. Of all the paintings offered for sale, the majority (74.22%) sold successfully. Oil paintings are by far the most common type of painting offered, at 74.06%. The other techniques in the sample are divided as follows: 17.53% water colors, 1.86% pastels, and 6.55% classified as mixed.

The natural logarithm of the hammer price of a painting is the dependent variable in our multivariate analysis. To facilitate comparison of prices across time, all prices are adjusted to 2020 prices using the consumer price index from Statistics Sweden as the deflator; prices are therefore expressed in real terms. The hammer price actually excludes the buyer's premium (which is added upon the hammer price), but the seller's commission is usually part of the hammer price (see [43]). In real terms, the most expensive painting sold in our sample is a water color by Anders Zorn ("Summer Fun") that sold for SEK 29.3 million (about $3.6 million) in June 2010 at Stockholms Auktionskammare. While such high-end sales receive a lot of media attention, the mean hammer price for paintings is much lower. Across all paintings in our sample, the average hammer price is SEK 215,400. The hammer price differs from the reservation price, thus emphasizing that sold paintings did achieve their reservation prices (the lowest price the seller would accept). The procedure in a standard auction is that the seller indicates a minimum price that the seller is willing to accept. If the highest bid is below this price, the painting is withdrawn or sometimes bought in by the auction house. See Beggs and Graddy [44] for a detailed discussion.

In Appendix B in S1 Appendix, we report distribution of hammer prices by painting technique. More specifically, this appendix shows prices by percentile, expressed in 2020 prices, along with the mean, median, standard deviation, skewness and kurtosis, as well as the lowest and highest values. Panel A reports hammer price sorted by technique and shows that the

**Table 1. Sample summary.**

|  | N | % |
|---|---|---|
| Number of paintings | 6,566 | |
| Number of artists | 416 | |
| Paintings sold | 4,873 | 74.22 |
| *Technique*: | | |
| Oil | 4,863 | 74.06 |
| Water color | 1,151 | 17.53 |
| Pastel | 122 | 1.86 |
| Mixed | 430 | 6.55 |
| *Sales characteristics*: | | |
| Auktionsverket | 2,684 | 40.88 |
| Bukowskis | 2,316 | 35.27 |
| Uppsala Auktionskammare | 1,566 | 23.85 |
| December (Yes = 1) | 3,328 | 50.69 |

[1] Sample descriptives regarding thematic classification are reported in Appendix D in S1 Appendix; this appendix shows the distribution along themes based on visual inspection and terminological analysis.

median hammer price is significantly lower than the mean, irrespective of technique, reflecting that data are highly positively skewed. Dispersion is also high because of outliers. Specifically, the highest standard deviation is for water colors, at more than four times higher than the mean. As reported in column (1), across all paintings, the mean (median) hammer price is SEK 215,400 (36,600). The mean price for a water color painting is SEK 370,600, and for an oil painting it is SEK 197,600. Of the two remaining painting techniques, the mean price is higher for pastel than for mixed. In summary, these data indicate that water colors are more expensive and have higher hammer price dispersion than oil paintings.

An important characteristic of a painting is its theme. Previous studies use the title of a painting to classify it into a theme and show that theme influences hammer price [4, 7]. To allow for comparison between the terminological approach and our visual inspection, we apply the approach used in Renneboog and Spaenjers [4] on our sample as well and test for differences. Renneboog and Spaenjers use the first word, or words, and a search string of the title in their classification into themes [4]. For instance, the theme "urban" can reflect a city, ville, town, village, street, rue, market, marche, harbor, port, Paris, London, Londres, New York, Amsterdam, Rome, Venice and Venise (see their Appendix B in S1 Appendix). We translated the keywords from Renneboog and Spaenjers into Swedish and then applied their approach.

Instead, our classification is based on visual inspection ("art-informed approach"). To this extent, each painting is first visually inspected and assigned to a theme with information gathered from auction catalogues available at the Art Library in Stockholm, or the home page of the auction house. If classification of a painting is unclear, we obtained further help with theme classification from art scholars at Uppsala University and the Art Library in Stockholm. Visual inspection of each painting took place between September 2019 and August 2021. We follow Garay [7] and classify a painting into one of the following eight themes: (1) animals, (2) landscape (and similar views), (3) nude, (4) object (e.g., ship, table), (5) people (excluding nude and portrait), (6) portrait, (7) still life (e.g., fruit, flowers, dead game, baskets) and (8) urban (e.g., city, street, market, harbor, port and buildings). Garay [7] also uses other themes, such as "Abstract", "Religion", "Self-portrait", "Untitled", and "Others", which are not part of the classification used. This is because there were no paintings belonging to the first three themes and those belonging to the last two could be classified based on the visual inspection. Finally, we obtained detailed data on hammer prices from the auction houses.

Our detailed classification procedure is likely to yield results that differ from those using a terminological approach. For example, the sample includes several paintings with titles such as "Sunrise," "Christmas," "Lena," "Summer," "Easter," and "Winter," which could cause classification problems if we had used a terminological approach, as there is a clear contrast between the title of the painting and what is actually shown, i.e., the theme of the painting. In addition, there are paintings without titles. In general, following discussions with Swedish art specialists, the fraction of untitled paintings can be as high as 15–20%, where a painting can undergo a change in title over time but also between auction houses.

To investigate whether visual inspection is relevant, we proceed with a robustness test, designed as follows. From our total sample, we randomly selected 10 paintings (see Appendix C in S1 Appendix) and made a document with a photograph of the painting and the name of the artist, but not the title of the painting; a sample of 10 paintings may seem small, but the goal here is simply to show that visual inspection delivers a more precise classification, compared to a terminological approach. Next, we asked one representative from each of the three auctions houses, five art scholars at Uppsala University, and three artists and teachers at the University of Arts, Crafts and Design in Stockholm, to classify the paintings into one of the eight themes. If a theme assigned by one respondent matched the theme assigned by our earlier visual inspection, one point would be given, zero otherwise. Thus, the total score could

vary from zero to 10 points for each respondent. We find that the average score for our respondents is 9.64, which is close to the maximum score. The score failed to reach 10 only because of a disagreement among the art specialists about the theme of one painting, a landscape by Carl-Fredrik Hill painted during his illness period (see painting A6 in Appendix C in S1 Appendix). Using a terminological approach, and similar to the keywords used in Renneboog and Spaenjers [4], yields an average score of 5.00 for 10 paintings. Accordingly, this test of theme classification provides evidence that our approach is substantially different from the terminological classification approach. However, we want to point out that it is much more time consuming as well.

Panel B in Appendix B in S1 Appendix shows the distribution of hammer prices by theme, where thematic classification was informed by visual inspection. The top three themes, based on the number of paintings, are landscape (31.6%), portrait (19.6%), and urban (14.1%). The theme with the smallest number of paintings is nudes (1.2%). This shows that paintings with this theme have the highest mean hammer price; the mean (median) hammer price for nudes is SEK 2,101,500 (360,000). Portraits are next with a mean (median) hammer price of SEK 319,600 (41,300). Object paintings rank third with a mean (median) hammer price of SEK 213,700 (34,200). Panel C shows the same when using the terminological approach. Here, the top three themes are no theme (44.0%), object (26.6%), and landscape (12.9%). The mean hammer price is highest for no theme, and the highest median observed is for animal.

Additionally, we include two variables to capture painting size. Here, we apply the same method as used elsewhere [18, 45–48]. That is, we include a surface variable to measure the surface in cm$^2$, and the square of surface. We expect to find a positive sign for the surface variable, and a negative sign for the squared variable. This concave nature of the surface area can be explained by the fact that while larger paintings are worth more, at some point, a painting can become too large, making it difficult to place or no longer manageable.

To control for various sales characteristics of a painting, we include an auction house dummy for each of the auction houses in our sample, and a dummy to denote whether the auction occurred in December. In general, we expect that the highest prices are for paintings sold at Bukowskis, as this generally is regarded as Sweden's leading auction house.

As part of the Heckman two-step approach, we include an exclusion restrictions in our model. This is the fraction of paintings sold at the preceding auction. Note that when using a lagged value, we would lose information from the first time period [49]. More specifically, we would lose the information related to the first set of three May-June auctions in 2010. However, the auction houses have provided an estimate of the fraction of sold paintings from the December auctions in 2009. This ensures that the information from the May-June auctions in 2010 is included in the analysis.

We present descriptive statistics for the variables in Table 2. This table shows that most of the paintings are by male artists, representing 87.2% of the sample. About 5% are unsigned, showing a high degree of authenticity in the sample. However, there is a caveat as signatures may be forged [50, 51]. Average painting size in our sample is 4,118 cm$^2$. Oil paintings are the most common type, at 74% of the sample. Landscapes are the most common theme, at 30.2%. The least common theme is nudes, at 1.2%. With the auction houses, Stockholms Auktionsverk accounts for the largest share of the sample, at 41%. Bukowskis sits at 35%, and Uppsala Auktionskammare has the smallest portion at 24%. We observe an almost even split between December auctions and May–June auctions. The average auction sells about three-quarters of the paintings put up for auction. Finally, we observe that the average artist has sold 65 paintings at previous auctions in the sample period. The last column (7) presents the results of a balancing test between the subsample of sold versus unsold paintings at their respective auctions. The balancing test is conducted as a linear regression, where each characteristic is regressed on

**Table 2. Hammer price descriptive statistics and balancing tests.**

| | (1) | (2) | (3) | (4) | (5) | (6) | (7) |
|---|---|---|---|---|---|---|---|
| | **Total** | | **Not sold** | | **Sold** | | |
| Variables | Mean | Std. dev. | Mean | Std. dev. | Mean | Std. dev. | Difference in means (6–4) |
| Price (SEK 000) | 215,4 | 996,0 | n/a | n/a | 215,4 | 996,0 | n/a |
| *Artist characteristics*: | | | | | | | |
| Unsigned (Yes = 1) | 0.052 | 0.222 | 0.056 | 0.229 | 0.050 | 0.219 | −0.005 |
| *Painting characteristics*: | | | | | | | |
| Surface (cm$^2$) | 4,118 | 4,948 | 4,516 | 5,226 | 3,979 | 4,841 | -536.373*** |
| Surface squared | 4.143e+07 | 1.697e+08 | 4.769e+07 | 1.424e+08 | 3.926e+07 | 1.782e+08 | -0.843e+07* |
| Mixed | 0.066 | 0.247 | 0.056 | 0.230 | 0.069 | 0.253 | 0.013* |
| Oil | 0.740 | 0.438 | 0.745 | 0.436 | 0.739 | 0.439 | −0.006 |
| Pastel | 0.019 | 0.135 | 0.027 | 0.161 | 0.016 | 0.125 | −0.011** |
| Water color | 0.175 | 0.380 | 0.172 | 0.377 | 0.176 | 0.381 | 0.005 |
| *Theme: Visual inspection* | | | | | | | |
| Landscape | 0.302 | 0.459 | 0.262 | 0.440 | 0.316 | 0.465 | 0.054*** |
| Animal | 0.133 | 0.339 | 0.122 | 0.328 | 0.139 | 0.343 | 0.014 |
| Nude | 0.012 | 0.108 | 0.011 | 0.105 | 0.012 | 0.109 | 0.001 |
| Object | 0.089 | 0.285 | 0.085 | 0.278 | 0.091 | 0.288 | 0.007 |
| People | 0.070 | 0.255 | 0.079 | 0.269 | 0.067 | 0.250 | −0.011 |
| Portrait | 0.211 | 0.408 | 0.255 | 0.436 | 0.196 | 0.397 | −0.059*** |
| Still life | 0.041 | 0.199 | 0.042 | 0.201 | 0.041 | 0.198 | −0.001 |
| Urban | 0.141 | 0.349 | 0.144 | 0.351 | 0.141 | 0.348 | −0.004 |
| *Theme: Terminological approach* | | | | | | | |
| Landscape | 0.066 | 0.248 | 0.061 | 0.240 | 0.068 | 0.251 | 0.011 |
| Animal | 0.171 | 0.376 | 0.162 | 0.369 | 0.174 | 0.379 | 0.006 |
| Nude | 0.004 | 0.064 | 0.004 | 0.064 | 0.004 | 0.064 | -0.000 |
| Object | 0.012 | 0.109 | 0.012 | 0.111 | 0.012 | 0.108 | -0.001 |
| People | 0.040 | 0.195 | 0.045 | 0.208 | 0.038 | 0.190 | -0.008 |
| Portrait | 0.025 | 0.156 | 0.025 | 0.157 | 0.025 | 0.155 | -0.001 |
| Still life | 0.029 | 0.168 | 0.031 | 0.174 | 0.028 | 0.166 | -0.003 |
| Urban | 0.061 | 0.238 | 0.063 | 0.243 | 0.060 | 0.237 | -0.004 |
| No theme | 0.593 | 0.491 | 0.594 | 0.491 | 0.592 | 0.491 | -0.002 |
| *Sales characteristics*: | | | | | | | |
| Auktionsverket | 0.408 | 0.492 | 0.477 | 0.500 | 0.385 | 0.487 | −0.092*** |
| Bukowskis | 0.353 | 0.478 | 0.315 | 0.465 | 0.366 | 0.482 | 0.050*** |
| Uppsala Auk. | 0.239 | 0.426 | 0.207 | 0.406 | 0.249 | 0.433 | 0.042*** |
| December (Yes = 1) | 0.493 | 0.500 | 0.511 | 0.500 | 0.487 | 0.500 | −0.024* |
| Fraction sold at previous auction | 0.757 | 0.0494 | 0.760 | 0.0530 | 0.755 | 0.0479 | -0.005*** |

Notes: Mean of each variable with standard deviation in parentheses. Column (7) is the result of a regression where each variable is regressed on a dummy for sold, equal to 1 if the painting was sold at auction, and 0 otherwise. All paintings are classified by visualization of each painting. Hammer prices are expressed in 2020 prices in Swedish Krona (SEK), where we use the consumer price index from Statistics Sweden as a deflator. Data reflect objects sold at three auction houses over the period 2010 to 2020. We use robust standard errors.

*** $p < 0.01$

** $p < 0.05$

* $p < 0.10$.

a dummy equal to 1 if the given painting was sold, and 0 otherwise. The advantage of applying a regression as a balancing test, over a simple t-test, is that with a regression, we are able to adjust the standard errors to Huber-White standard errors which are robust to the presence of heteroscedasticity. The balancing test shows persistent differences between the two subsamples, signaling the presence of sample selection bias. In particular, we find statistically significant differences between sold and unsold pieces with respect to the painting's size and the use of pastel: larger paintings are sold less, as are pastels. Furthermore, with theme, there are significant differences with landscapes and portraits: landscapes are sold more, portraits less so. We want to highlight that there are no significant differences between sold and unsold paintings with the terminological-based thematic classifications.

## 3. Results

This section reports model estimation results and discusses our findings.

### 3.1 Price drivers of Swedish classical paintings

Table 3 shows the parameter estimates of our model. Recall that this is based on the Heckman model, which is an improvement over the traditional hedonic regression approach as it takes into account the possibility that a painting will not sell when put up for auction. It is reasonable to believe that paintings sold at auction may differ from those that do not sell. We already observe such differences in our sample in the balancing tests in Table 2. An OLS estimation on a (non-random) sample of sold paintings would produce biased and inconsistent estimates. This sample selection bias is created because our dependent variable is observed only when a painting is sold (see [32]). That is, we observe the (log) hammer price only if the dummy variable indicating whether a painting is sold is equal to 1. This motivates use of the Heckman approach as described in section 2.

Table 3 presents the results from estimating the Heckman model. This two-step model is estimated using maximum likelihood, where the dependent variable in the first step regression is a dummy variable equal to 1 if a painting was sold at auction, and 0 otherwise. The dependent variable in the second step price regression is the log of the real hammer price in SEK. In this setup, the second step price equation can be viewed as a hedonic regression, where any potential sample selection bias is controlled for via inclusion of the additional regression, the IMR. The result for the selection equation is reported in column (3), and the results of the second step price equation are shown in column (1), with the corresponding transformation into percentage impact on price in column (2). The selection parameter known as the IMR (denoted as lambda in the table) is shown in column (1) as well. The statistical significance of this parameter and that of the exclusion restriction indicates that there is evidence of a sample selection mechanism. More specifically, the positive coefficient of the IMR suggests that paintings that do sell are sold for a higher price than what is implied by their characteristics.

When determining the probability of a painting being sold, reported in column (3), it turns out that painting characteristics, namely size, technique (pastel), and theme (nude, people), and sales characteristic (auction house, timing) significantly matter. We observe that larger paintings are generally harder to sell, given the negative sign of the surface variable and the positive sign of the squared surface term. However, the effect is rather small, as the coefficients are zero at three decimal places. Regarding painting technique, only using pastel seems to influence the probability of being sold at auction. Compared to oil paintings, pastels are significantly less likely to sell. In addition, the themes nude and people, and portraits have a significant impact on the probability of being sold (compared to landscapes): Nudes are sold much easier and portraits harder so. Furthermore, the auction house has an important impact on

**Table 3. Price impact on Swedish classical paintings.**

| Variable | (1) Estimated coefficient–Heckman | (2) Estimated price impact (%)–Heckman | (3) First-stage Probit–Heckman |
|---|---|---|---|
| *Artist characteristics*: | | | |
| Unsigned (Yes = 1) | -0.410*** | -33.63 | -0.147 |
| | (0.093) | | (0.090) |
| *Painting characteristics*: | | | |
| *Size of painting* | | | |
| Surface (cm$^2$) | 0.000*** | 0.01 | -0.000*** |
| | (0.000) | | (0.000) |
| Surface squared | -0.000*** | -0.00 | 0.000** |
| | (0.000) | | (0.000) |
| *Technique (Base type: Oil)* | | | |
| Mixed | -1.441*** | -76.33 | -0.073 |
| | (0.098) | | (0.104) |
| Pastel | -1.027*** | -64.19 | -0.376*** |
| | (0.142) | | (0.135) |
| Water color | -0.264*** | -23.20 | -0.022 |
| | (0.076) | | (0.078) |
| *Theme (Base theme: Landscape)* | | | |
| Animal | 0.125 | 13.31 | -0.016 |
| | (0.081) | | (0.084) |
| Nude | 0.432** | 54.03 | 0.363* |
| | (0.197) | | (0.208) |
| Object | 0.004 | 0.40 | -0.033 |
| | (0.059) | | (0.066) |
| People | -0.003 | -0.30 | -0.164** |
| | (0.071) | | (0.074) |
| Portrait | 0.079 | 8.22 | -0.074 |
| | (0.061) | | (0.063) |
| Still life | -0.146 | -13.58 | -0.063 |
| | (0.148) | | (0.165) |
| Urban | 0.134*** | 14.34 | 0.024 |
| | (0.050) | | (0.055) |
| *Sales characteristics*: | | | |
| *Auction house (base Auktionsverket)* | | | |
| Bukowskis | 0.389*** | 47.55 | 0.155*** |
| | (0.036) | | (0.039) |
| Uppsala Auktionskammare | -0.106*** | -10.06 | 0.220*** |
| | (0.039) | | (0.044) |
| *Winter sale dummy*: | | | |
| December | 0.013 | 1.31 | -0.064** |
| | (0.029) | | (0.032) |
| *Exclusion restrictions and selection parameter*: | | | |
| Fraction sold at previous auction | | | -0.643** |
| | | | (0.327) |
| Lambda | 1.033*** | | |
| | (0.023) | | |
| Constant | 2.527*** | | 1.088*** |

(*Continued*)

**Table 3.** (Continued)

| Variable | (1) Estimated coefficient–Heckman | (2) Estimated price impact (%)–Heckman | (3) First-stage Probit–Heckman |
|---|---|---|---|
| | (0.313) | | (0.395) |
| Artist fixed effects | Yes | | Yes |
| Year fixed effects | Yes | | Yes |
| Observations | 6,566 | | 6,566 |

Notes: The dependent variable is the log of hammer price in SEK at 2020 price levels for column (1). The dependent variable for column (3) is a dummy variable equal to 1 if the painting was sold at auction, and 0 otherwise. The model is estimated via Maximum Likelihood. Lambda refers to the inverse Mills ratio. Artist fixed effects and Year dummies are included in estimation of the regression models, but not reported. Due to the large number of artists in our sample, artist with less than 10 paintings in our sample are grouped together as "Other artists". Robust standard errors in parentheses

*** $p < 0.01$

** $p < 0.05$

* $p < 0.10$.

whether a painting will sell, with paintings more likely to sell at Bukowskis and Uppsala Auktionskammare compared to Auktionsverket. Finally, we observe that paintings at December auctions are less likely to sell.

The Heckman procedure confirms that the exclusion restriction carries explanatory power when it comes to explaining the probability of a painting being sold at auction. Specifically, we observe that if a larger fraction of paintings was sold at the preceding auction, the probability of sale declines. One possible explanation is that if a larger fraction of paintings were sold at the preceding auction, it might indicate a weaker or oversaturated art market. Buyers may be more cautious in subsequent auctions, leading to a decline in the probability of sale as they have more options and potentially less urgency to make purchases. We evaluate the correlation between the IMR and our exclusion restriction to better understand the strength of our chosen exclusion restriction. We find a correlation coefficient of 0.11. This relatively low correlation combined with our large sample size indicates that we have strong exclusion restrictions [33].

For artist characteristics, we find no significant difference between paintings by male and female artists respectively. This is not in line with some other studies [52–54]. Furthermore, unsigned works deliver on average a 34% discount compared to signed work.

Inspection of painting characteristics shows that larger paintings are generally more expensive, but only until they reach an inflection point at 77625.9 cm$^2$ (7.76m$^2$) The oils are most precious as they are skill intensive. Paintings using water colors, pastel or a mixed technique receive a lower hammer price, compared to oil paintings, by 23%, 64% and 76%, respectively. This is in line with previous literature, namely that oil paintings are more expensive than paintings using other media [4, 18]. For theme, we observe that paintings with animals, nudes, and portraits all have a substantial higher hammer price compared to landscapes; the opposite is the case with a still life. It is interesting that animal paintings, while carrying a premium on hammer price, are harder to sell.

The coefficients related to painting sales characteristics show the following: Paintings sold at Bukowskis sell at a premium of 48% compared to Auktionsverket. Combining this with the findings from the selection equation, paintings not only sell more easily at Bukowskis, they also have a much higher hammer price. This differs for Uppsala Auktionskammare. Paintings are estimated to receive a lower hammer price of about 10% compared to Auktionsverket. However, as is evident from the selection equation, the paintings are on average easier to sell compared to Auktionsverket.

We want to find out how informative our analysis is using visual inspection to classify paintings along themes. To this extent, we used the terminological approach of Renneboog and Spaenjers [4] to classify the sample paintings along 'terminological themes', as discussed in section 2. We have reported the distribution of paintings across the different theme in Appendix D in S1 Appendix. Please note that when using the terminological approach, a painting can also be assigned 'No theme' if the method is unsuccessful in allocating the painting to one of the other themes.

We use the same Heckman model and compare the estimation results from 'visual inspection' with this 'terminological approach'. These results are in Table 4 and we concentrate on the comparison of columns (1) and (4).

First, we want to highlight that the number of observations used to arrive at this table is reduced dramatically compared with Table 3. This is due to the fact that the terminological approach of Renneboog and Spaenjers renders about 60% of the paintings as 'unclassified'. In the comparison, we only compare the paintings that are classified in the terminological analysis. Therefore, columns (1)-(3) in Table 4 show different results compared to columns (1)-(3) in Table 3. We test whether there is a systematic bias because of dropping so many paintings (see Appendix F in S1 Appendix). It occurs that this indeed is the case with many features: Gender, size, technique (water color), theme (nude, object, people, portrait, urban), and auction house (Bukowskis). This systematic bias suggests that thematic classification based on the terminological analysis results in measurement error, as many paintings are (mistakenly) assigned as 'unclassified'.

Table 4 shows that the lambda (related to the Inverse Mills ratio, IMR) with visual inspection (column 1) is highly comparable with that in Table 3. However, it is highly different from the lambda reported in column (4). The latter is not statistically significant. This implies that there is no selection effect, meaning that there is no evidence of a sample selection by using terminological classification. This motivates the case for employing visual inspection over semantics.

Table 4 also shows that there are no substantial differences regarding the estimations for the influence of artist characteristics, painting size, painting technique, and sales characteristics. However, and highly interesting, there are substantial differences between the thematic characteristics of the paintings along visual versus terminological analysis. In particular, this concerns the themes animal, nude, object, and people. As an example, in Table 4 the coefficient for 'nude' in column (1) is negative and not significant, in contrast to that in Table 3. We explain this by the observation that especially paintings in this category have strange names. Recall Fig 1: Here, 'Broken Mirror' and 'Red Sand' where being labelled as 'unclassified' in the terminological analysis, whereas visual inspection suggests 'nude'. Furthermore, see Appendix D in S1 Appendix, the fraction of paintings classified as 'nude' with terminological analysis is much lower than that based on visual inspection. We do not have the competence to explain why especially nudes are so hard to classify with the terminological analysis. However, this clearly indicates the difficulty in using the terminological approach. It is interesting to note that we observe an insignificant IMR in column (4). However, this does not always imply that sample selection bias is not present. Simulation results reported by Trevis Certo et al. [33] indicate that a weak exclusion restriction and/or small sample may result in an insignificant IMR, even in the presence of sample selection bias. Given a loss of roughly 60% of our sample between Tables 3 and 4, it seems plausible that a (much) smaller sample size is responsible.

**Table 4. Price impact on Swedish classical paintings–visual vs terminological approach.**

| | (1) | (2) | (3) | (4) | (5) | (6) |
|---|---|---|---|---|---|---|
| | Visual inspection | | | Terminological approach | | |
| Variable | Estimated coefficient–Heckman | Estimated price impact (%)–Heckman | First-stage Probit–Heckman | Estimated coefficient–Heckman | Estimated price impact (%)–Heckman | First-stage Probit–Heckman |
| *Artist characteristics*: | | | | | | |
| Unsigned (Yes = 1) | -0.451*** | -33.30 | 0.132 | -0.508*** | -39.83 | -0.015 |
| | (0.141) | | (0.143) | (0.120) | | (0.128) |
| *Painting characteristics*: | | | | | | |
| *Size of painting* | | | | | | |
| Surface (cm$^2$) | 0.000*** | 0.01 | -0.000 | 0.000*** | 0.01 | -0.000 |
| | (0.000) | | (0.000) | (0.000) | | (0.000) |
| Surface squared | -0.000*** | 0.00 | 0.000 | -0.000*** | 0.00 | 0.000 |
| | (0.000) | | (0.000) | (0.000) | | (0.000) |
| *Technique (Base type: Oil)* | | | | | | |
| Mixed | -1.026*** | -64.16 | 0.211 | -1.181*** | -69.30 | 0.212 |
| | (0.150) | | (0.134) | (0.148) | | (0.141) |
| Pastel | -0.670*** | -48.83 | -0.180 | -0.526** | -40.90 | -0.223 |
| | (0.225) | | (0.226) | (0.222) | | (0.214) |
| Water color | -0.231** | -20.63 | 0.132 | -0.236** | -21.02 | 0.022 |
| | (0.110) | | (0.092) | (0.103) | | (0.090) |
| *Theme (Base theme: Landscape)* | | | | | | |
| Animal | 0.259** | 29.56 | -0.044 | 0.089 | 9.31 | -0.004 |
| | (0.110) | | (0.069) | (0.085) | | (0.079) |
| Nude | -0.165 | -15.21 | 0.644** | -0.508* | -39.83 | 0.092 |
| | (0.277) | | (0.320) | (0.276) | | (0.262) |
| Object | 0.031 | 3.15 | -0.167 | -0.178** | -16.31 | -0.023 |
| | (0.091) | | (0.112) | (0.089) | | (0.161) |
| People | 0.200 | 22.14 | -0.069 | 0.091 | 9.53 | -0.148 |
| | (0.131) | | (0.146) | (0.083) | | (0.095) |
| Portrait | 0.060 | 6.18 | -0.029 | -0.022 | -2.18 | -0.067 |
| | (0.084) | | (0.082) | (0.109) | | (0.118) |
| Still life | 0.090 | 9.42 | -0.122 | 0.063 | 6.50 | -0.072 |
| | (0.140) | | (0.097) | (0.146) | | (0.108) |
| Urban | 0.101 | 10.63 | -0.118 | 0.132* | 14.11 | -0.047 |
| | (0.076) | | (0.081) | (0.069) | | (0.082) |
| *Sales characteristics*: | | | | | | |
| *Auction house (base Auktionsverket)* | | | | | | |
| Bukowskis | 0.392*** | 47.99 | 0.302*** | 0.206*** | 22.88 | 0.379*** |
| | (0.049) | | (0.060) | (0.043) | | (0.064) |
| Uppsala Auktionskammare | -0.109** | -10.33 | 0.225*** | -0.248*** | -21.96 | 0.277*** |
| | (0.054) | | (0.066) | (0.047) | | (0.071) |
| *Winter sale dummy*: | | | | | | |
| December | 0.013 | | -0.093* | 0.058 | | -0.076 |
| | (0.041) | | (0.050) | (0.036) | | (0.053) |
| *Exclusion restrictions and selection parameter*: | | | | | | |

*(Continued)*

**Table 4.** (Continued)

| | (1) | (2) | (3) | (4) | (5) | (6) |
|---|---|---|---|---|---|---|
| | **Visual inspection** | | | **Terminological approach** | | |
| **Variable** | **Estimated coefficient–Heckman** | **Estimated price impact (%)–Heckman** | **First-stage Probit–Heckman** | **Estimated coefficient–Heckman** | **Estimated price impact (%)–Heckman** | **First-stage Probit–Heckman** |
| Fraction sold at previous auction | | | -0.641 | | | -0.764 |
| | | | (0.533) | | | (0.740) |
| Lambda | 0.880*** | | | -0.025 | | |
| | (0.034) | | | (0.052) | | |
| Constant | 2.694*** | | 1.058*** | 2.949*** | | 1.210** |
| | (0.161) | | (0.385) | (0.169) | | (0.525) |
| Artist fixed effects | Yes | | Yes | Yes | | Yes |
| Year fixed effects | Yes | | Yes | Yes | | Yes |
| Observations | 2,673 | | 2,673 | 2,673 | | 2,673 |

Notes: Columns (1)-(3) applies a thematic classification of the paintings following a visual inspection. Columns (4)-(6) applies a thematic classification of the paintings following a terminological approach as in Renneboog and Spaenjers [4]. The number of observations in the estimations presented in this table are lower compared to the ones in Table 3, as paintings that did not get a thematic classification following the terminological approach were dropped from the estimation sample. This is done as to better compare the results following the two approaches to thematic classification. The dependent variable is the log of hammer price in SEK at 2020 price levels for columns (1) and (4). The dependent variable for column (3) and (6) is a dummy variable equal to 1 if the painting was sold at auction, and 0 otherwise. The models are estimated via Maximum Likelihood. Lambda refers to the inverse Mills ratio. Artist fixed effects and Year dummies are included in estimation of the regression models, but not reported. Due to the large number of artists in our sample, artist with less than 10 paintings in our sample are grouped together as "Other artists". Robust standard errors in parentheses

*** p<0.01

** p<0.05

* p<0.10.

## 4. Conclusion

We study the pricing attributes of paintings using a hand-collected data set for Swedish Masters in an effort to address two material issues overlooked in prior studies: (a) classification of a painting into an artistic theme; and (b) sample selection bias due to the paintings not sold. Prior studies typically infer a painting's theme by screening its title and use samples comprising only paintings that actually sold at auction. Our approach allows for the examination of the pricing of paintings in greater detail using more precise data, thereby providing novel evidence regarding the pricing mechanism. We argue that visual inspection captures the theme better than relying on the name of the painting, but is more time-consuming. Our more precise classification approach reduces measurement error and improves internal validity of the results.

Using the Heckman two-stage procedure for a sample of 6,566 paintings by Swedish Masters applied to data from the three major Swedish auction houses from 2010 to 2020, we find significant differences in painting characteristics between sold and unsold paintings. We establish that size, technique, theme, and sales characteristics all can have a significant impact on the paintings' price. In particular, smaller sized artworks, oil paintings, nudes, urban, Bukowski auction, yield (relatively) higher prices. We compare the visual classification of the themes of the paintings with terminological classification. Here, we find that the latter is unable to classify about 60% of the sample paintings, in particular the category 'nude'. We

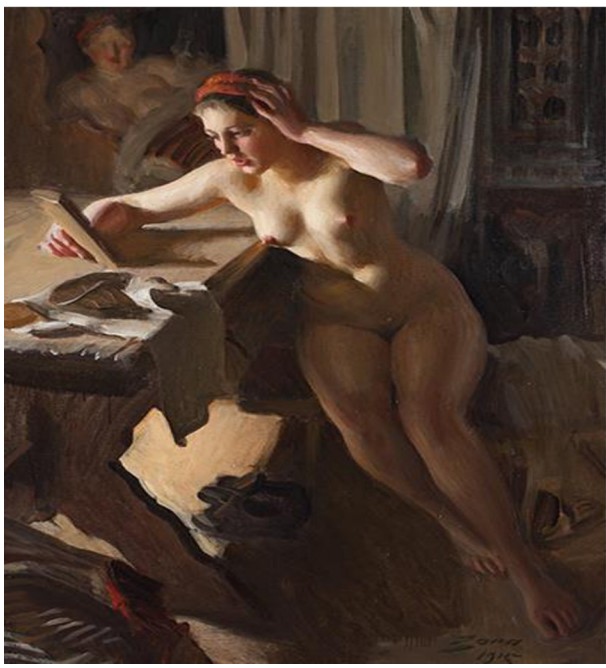 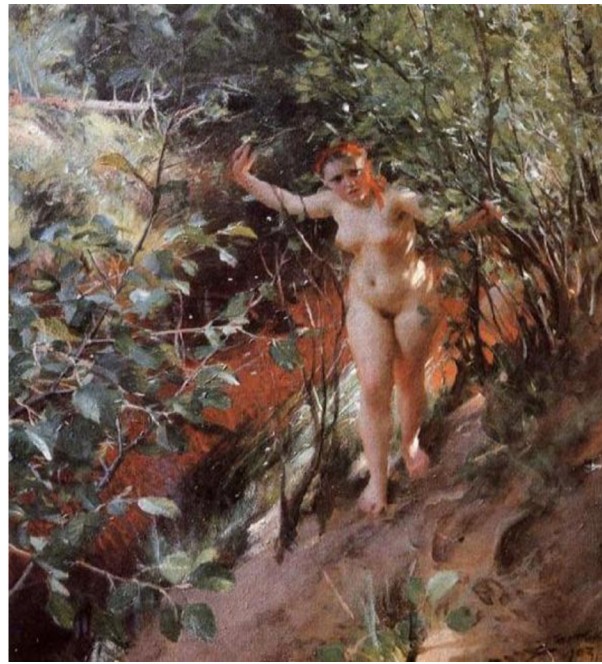

**Fig 1. What is in a Name?** Left painting: Oil painting by Anders Zorn is "Old Mirror" (Swedish: "Gammal spegel"). Visual inspection results in classifying it in the theme "Nude", whereas the terminological labeled it as 'unclassified'. The painting was sold at Bukowskis in 2020 for SEK million 3.6 (excl. commission). Right painting: Oil painting by Anders Zorn is "Red Sand" (Swedish: "Röd sand"). Visual inspection results in classifying it in the theme "Nude", whereas the terminological labeled it as 'unclassified' The painting was sold at Bukowskis in 2010 for SEK million 11.8 (excl. commission). Both photographs are from Bukowskis, who provided a CC BY license.

show there is a systematic bias when we employ a terminological analysis, suggesting measurement error.

This study contributes to the rich empirical literature on the arts and the factors that affect the prices of paintings using a large, hand-collected dataset with detailed characteristics for each painting. We offer an explanation as to why our classification approach is superior compared to the conventional approach based on the names of paintings, and it addresses sample selection bias. We also want to point out that our innovative approach very well aligns with the practices commonly used in auction houses and galleries when estimating the value of artwork. In future work, we hope to use machine learning to carry out classification of paintings based on the artwork's properties. We strongly believe that the literature can benefit from such an approach. In addition, and given the significant dispersion in hammer prices across paintings, a relevant follow-up question is whether there is a difference in price elasticity between inexpensive and expensive paintings. Further, the highly concentrated number of auction houses in Sweden raises the question of whether the art market is efficient. In future work, we intend to analyze the link between hammer prices and the level of auction house concentration using the Herfindahl-index as a proxy for concentration.

## Supporting information

**S1 Data. Consists of data and codes for this submission.**
(DTA)

**S1 Appendix. Consists of the supporting files for this submission.**
(DOCX)

## Acknowledgments

We appreciate insightful comments from Maria Brunskog. We are also grateful for helpful comments from Susanna Carlsten, Johan Cederlund, Britt-Inger Johansson, Knut Knutsson, Emely Stammers, Greger Sundin, Kristina Vinell, Lars Vinell, Lars Wängdal, and scholars and art specialists who participated in our robustness test. We also want to thank Simon Porcher, editor, and anonymous reviewers of PLOS ONE for their comments and constructive suggestions. Especially "reviewer 1" has motivated us to pursue a systematic comparison between artistic and terminological analysis and has been a very kind guide to this field.

## Author Contributions

**Conceptualization:** Adri De Ridder, Bert Scholtens.

**Formal analysis:** Adri De Ridder, Steffen Eriksen.

**Investigation:** Adri De Ridder, Steffen Eriksen, Bert Scholtens.

**Methodology:** Adri De Ridder, Steffen Eriksen.

**Project administration:** Bert Scholtens.

**Supervision:** Bert Scholtens.

**Validation:** Adri De Ridder, Steffen Eriksen.

**Visualization:** Steffen Eriksen.

**Writing – original draft:** Adri De Ridder.

**Writing – review & editing:** Steffen Eriksen, Bert Scholtens.

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
