## [Decision Letter · Decision Letter 0]

15 Jun 2023

PONE-D-23-12660The Art of Valuation: Incorporating Expert Insights to Price Classical Paintings by Swedish Masters *PLOS ONE

Dear Dr. Scholtens,

Thank you for submitting your manuscript to PLOS ONE. After careful consideration, we feel that it has merit but does not fully meet PLOS ONE’s publication criteria as it currently stands. Therefore, we invite you to submit a revised version of the manuscript that addresses the points raised during the review process.

We look forward to receiving your revised manuscript.

Kind regards,

Simon Porcher

Academic Editor

PLOS ONE

Journal Requirements:

2. Thank you for stating the following financial disclosure: "No specific funding was received for this work"

4. Please include a separate caption for figure 1 in your manuscript.

5. We note that Figure 1 and Supporting Figures A1 to A14 in your submission contain copyrighted images. All PLOS content is published under the Creative Commons Attribution License (CC BY 4.0), which means that the manuscript, images, and Supporting Information files will be freely available online, and any third party is permitted to access, download, copy, distribute, and use these materials in any way, even commercially, with proper attribution. For more information, see our copyright guidelines: http://journals.plos.org/plosone/s/licenses-and-copyright.

(1) You may seek permission from the original copyright holder of Figure 1 and Supporting Figures A1 to A14 to publish the content specifically under the CC BY 4.0 license. 

(2) If you are unable to obtain permission from the original copyright holder to publish these figures under the CC BY 4.0 license or if the copyright holder’s requirements are incompatible with the CC BY 4.0 license, please either i) remove the figure or ii) supply a replacement figure that complies with the CC BY 4.0 license. 

Please check copyright information on all replacement figures and update the figure caption with source information. 

If applicable, please specify in the figure caption text when a figure is similar but not identical to the original image and is therefore for illustrative purposes only.

**Additional Editor Comments:**

Reviewer 1 particularly highlight some major comments - e.g. on some claims, or selection effect in the sample -you should deal with. 

Reviewers' comments:

Reviewer's Responses to Questions

**Comments to the Author**

1. Is the manuscript technically sound, and do the data support the conclusions?

Reviewer #1: Partly

Reviewer #2: Yes

2. Has the statistical analysis been performed appropriately and rigorously? 

Reviewer #1: Yes

Reviewer #2: Yes

3. Have the authors made all data underlying the findings in their manuscript fully available?

Reviewer #1: Yes

Reviewer #2: Yes

4. Is the manuscript presented in an intelligible fashion and written in standard English?

Reviewer #1: Yes

Reviewer #2: Yes

5. Review Comments to the Author

Reviewer #1: The paper “The art of valuation: incorporating expert insights to price classical paintings by Swedish masters” focuses on an important point regarding artworks’ evaluation: the theme of the artwork. This characteristic is likely to be crucial in how collectors and other market agents evaluate artworks on sale in the market, and previous studies have investigated this issue. However, as rightly stated by the authors, it must be stressed that themes are not always easy to identify unless a visual inspection of the artwork is made. Moreover, this is not always the case, as expertise is sometimes required to accurately attribute a theme to an artwork. As a motivation for this study, I find that the authors are raising an intriguing point, contributing to the cultural economics analysis on the topic.

However, I think that the paper has some issues that should be considered. In what follows, I will outline a series of points that I believe should be considered by the authors.

1) While, as stated above, the point raised by the authors is relevant, the results and discussion solely focus on the information presented in the table, confining the analysis to the statistical significance of the coefficients alone and to the test for the sample selection. Although there are undoubtedly important, I perceive a gap between the study's objective and the ensuing discussion. In the discussion, the authors should explicitly emphasize that: i) the theme of the artworks significantly influences their value, as established in prior literature, and ii) expert inspection should take precedence over terminological analysis based on the artwork’s title. While the first claim is already possible with the results contained in the paper, the same cannot be said for the second claim. In fact, the paper lacks a comparison of the outcomes produced by the 'terminological approach. To truly highlight the significance of expert inspection in art market evaluation, I believe the paper should incorporate this comparison, otherwise a significant disparity exists between the research question and the analysis. This comparison could also suggest the existence of an impact of measurement error on the estimated coefficients. As for now, there is no proof of an impact of this measurement error. In fact, the comparison between the two methods is only made in the inspection of the 10 paintings the authors report in the appendix, but this does not directly relate to the prices associated with specific themes in the market: this ex-ante evaluation is made outside the market and, while it supports the claim that expert inspection could be preferred to terminological deduction of the theme based on the title (although it is only based in few data points), in order to mitigate measurement error, it could not be extended to the impact on the prices.

Making the suggested comparison between the results obtained using the two methods (and clarifying that the ex-ante evaluation is not directly linked to the prices) would bring greater focus to the authors' suggested innovation, which aligns with the approach commonly used in auction houses and galleries when estimating the value of artwork. This last point should be raised as well.

2) The authors state (p. 12, line 266 and following) that they make use of Garay’s (2021) classification for their themes. Indeed, Garay also makes use of other themes, such as “Abstract”, “Others”, “Religion”, “Self-portrait”, and “Untitled”, which are not part of the classification used by the authors; did the authors drop the paintings falling into these 5 classes? Or it is the case that any of the art pieces analyzed fell into any of these classes? This should be clarified.

Further, the authors’ point (7) called “Still life” also contain animals, which are also part of point (1): this can lead to confusion (see page 12, line 269: “still life (e.g., flowers, animals, food)”.

Another possible issue related to this classification (and Garay’s as well) is that I points (3) “nude”, (5) “people”, and (6) “portrait” are all actually about people. Point (5) should then be “people” excluding “nude” and “portrait”, otherwise an artwork of a nude portrait (Anders Zorn made nude portraits and is likely to be part of the used dataset, being other of his paintings in this paper’s Figure 1) could end up in any of the three groups. The authors should explicitly state the contents of each chosen class to enhance clarity and ensure replicability of the analysis. Without this information, it is difficult to understand how the analysis was conducted and replicate the study. Additionally, it would be valuable for the authors to include the perspective of the experts who conducted the visual inspection regarding the classification employed in the paper. This would provide valuable insights and further strengthen the study.

3) Concerning the inspection, the title and the abstract seem to suggest that this is made by experts and art specialists. However, only a portion of the inspection was conducted by these experts. Some of the artwork themes may be easily identifiable even without advanced expertise, but I am wondering who actually made the inspection of the pieces not examined by the experts. Were the authors responsible for the inspection? If so, how was it conducted? If multiple individuals conducted the inspection for pieces not examined by the experts, did they perform it independently and compare their chosen themes for consistency? How many pieces were inspected by the experts and how many by the others? All this information should be shared.

Moreover, it is important to highlight in the abstract and introduction that not all inspections were conducted by experts. Furthermore, it is inaccurate to claim that the experts improve the classification compared to the terminological approach if some of the inspections were conducted by non-experts.

4) Concerning the methodology used (hedonic OLS regression and Heckman’s model), I do not think there is need to report OLS results in the paper, since the authors find that a sample selection bias exists. I think the OLS results could be removed or, at most, relegated to the appendix.

5) The authors state that they are following recent literature and introduce the artist’s characteristics in their analysis. This is actually not so recent (see for example Nahm 2010, Anderson et al. 2016). What is more recent is the use of artist fixed effects to capture the artist’s name effect on his/her artwork price (Cleeremans et al. 2016, Hernando and Campo 2017, Radermecker 2019, Angelini et al. 2023). I think the authors should also add this fixed effect to their analysis, since previous studies found that this effect exists, and it is also likely to influence the theme effect the authors are investigating (e.g., a nude by Zorn is likely to be sold at a higher price than a nude by Eugène Jansson).

6) Since this paper mostly falls and relates to cultural economics literature, the authors should expand the coverage of previous papers in this strand of literature. In fact, the analysis of themes and topics (sometimes referred to as "genre") is not new (e.g., Bocart & Oosterlinck, 2011, among many others, besides the other papers already cited by the authors), and the same is true for the use of the Heckman model, mostly after Collins et al. (2009) paper. See for example Bocart and Oosterlinck (2011), Ekelund et al. (2013), and Angelini et al. (2022). Some scholars also focused on the importance of the title (e.g., Park et al. 2022, Russell and Milne 1997). Recent works (Tan et al. 2016; Sheppard 2021; Choi et al. 2023) also implement what the authors suggest in their future extensions (or something along these lines), namely “use artificial intelligence” (p. 21, line 495).

In addition to that, some of the claims of the authors are inaccurate and might need additional support from previous studies or additional information. I list these in the following:

a. Page 8, lines 178-180: “This is because artists that have previously sold a painting are more likely to sell again at an upcoming auction.” This period needs a supporting reference or a source.

b. Page 10, lines 221-222: “We exclude a small number of paintings withdrawn from auctions prior to the event when their provenance was not clear.” What a “non clear” provenance means should be explained. Is it that the published provenance in the catalogue was then deemed to be wrong? Or else? Also, how many paintings have been excluded for this reason should be stated. The same should be stated for the other reasons of exclusion (p. 9, line 211 to p. 10 line 214).

c. Page 10, line 235: “all auction prices exclude fees”. The hammer price actually excludes the buyer’s premium (which is added upon the hammer price), but the seller’s commission is usually part of the hammer price. See Ashenfelter and Graddy (2003).

d. Page 11, line 244: “When a painting is bought by an auction house, its future price may be affected”. It is not clear what this means.

e. Page 15, line 358: the authors refer to the signature as a proof of authenticity. Forged artworks are an issue in the art market, and a lot of these are signed (with a forged signature). See for example Vuong et al. (2018) and Euwe and Oosterlinck (2017). A proof of authenticity would be a certificate of authenticity, not the signature (so the period at page 16, lines 360-361 should be modified).

f. Page 4, line 87-88: “paintings by male artists sell at considerably higher prices than those from female artists”. Adams et al. (2021), already cited in the paper, should be recalled in the comparison. Other papers to use in the comparison are Cameron et al. (2019) and LeBlanc and Sheppard (2022).

7) The authors claim that the Swedish art market can be considered a representative sample of the general art market. However, I have reservations about this assertion, as the art market is known for its significant diversity. Furhtermore, if we limit the comparison to the average selling prices, in this paper this is 215,400 SEK (around 20,000$), while the average price in other papers is way higher (see for example the average of prices in Collins et al. (2009), 223,954.800$, or the one in Renneboog and Spanjers (2013), 159,354$; both papers focus on sales in the most important international auction houses). In fact, the Swedish market is not comparable to the New York, the London, or the Paris market. I believe the authors should reconsider the extent of their claim regarding the generalizability of their analysis. While this adjustment does not diminish the research question's interest, asserting that their results apply to the entire art market is inaccurate.

Some additional minor points (which need anyway to be clarified) follows:

A) The authors find some unconventional results in their analysis, which should be discussed (and compared to previous results). Examples are:

a. watercolors fetch on average a higher price in the Swedish art market. Could it be that only very high-quality watercolors enter the market represented by the three auction houses the authors are considering, while lower-level watercolors are relegated to other auction houses or dealers?

b. As a larger fraction of paintings was sold at the preceding auction, the probability of sale declines (p. 19, lines 432-433). What could be an explanation of this?

B) The authors claim at page 4, lines 92-93, that one of their novel contributions is that they provide detailed evidence that differences exist between sold and unsold pieces. This is actually something that is implicated by any previous results in the literature using the Heckman model and finding that a sample selection issue is at work (see point 6 above). The claim should be modified.

C) Concerning the period “Some issues with the hedonic approach include assumptions that there are no changes in the parameters across the sample period, and that it follows an implicit functional form” (p. 5, lines 111-112), it is not clear what “it” in “it follows” refers to.

D) At p. 6, line 126, the authors state that the dependent variable of equation (1) is the price, but it is actually the log of the price. Same issue at page 10, line 232.

E) At p. 9, lines 205-208, the authors state that online databases have a series of issues that motivated them to not choosing these, and that these databases have “problematic” classification of paintings. Both claims should be better explained, since they are quite vague.

F) It would be better to keep the same unit of measurement for the surface throughout the paper. Sometimes m^2 are used, some other cm^2.

G) At page 21, line 489, the authors report the claim “sex sells” as an explanation of their result related to the nude premium. However, nude in art and sex are far to be the same thing, unless one expects all nude art to be sexualized (which I think is quite unlikely). A wide discussion on this point can be found in Nead (1992).

H) In table 1, the authors sum the number of artists per year (column 3). I do not think summing these numbers make sense, since some artists have surely been exchanged in more than one year.

References

Anderson, S.C., Ekelund, R.B., Jackson, J.D. and Tollison, R.D. (2016). Investment in early American art: the impact of transaction costs and no-sales on returns. J Cult Econ 40, 335–357.

Angelini, F., Castellani, M. & Pattitoni, P. (2022). You can’t export that! Export ban for modern and contemporary Italian art. Eur J Law Econ. https://doi.org/10.1007/s10657-022-09759-0

Angelini, F., Castellani, M., & Pattitoni, P. (2023). Artist Names as Human Brands: Brand Determinants, Creation and co-Creation Mechanisms. Empirical Studies of the Arts, 41(1), 80–107.

Ashenfelter, Orley and Kathryn Graddy. 2003. "Auctions and the Price of Art." Journal of Economic Literature, 41 (3): 763-787.

Bocart, F. and Kim Oosterlinck (2011). Discoveries of fakes: Their impact on the art market. Economics Letters, Volume 113, Issue 2, Pages 124-126.

Cameron, L., Goetzmann, W.N. & Nozari, M. (2019). Art and gender: market bias or selection bias?. J Cult Econ 43, 279–307.

Choi, J., Lan Ju, Jian Li, and Zhiyong Tu (2023). Information extraction and artwork pricing. arXiv:2302.08167

Cleeremans, A., Ginsburgh, V., Klein, O., & Noury, A. (2016). What’s in a Name? The Effect of an Artist’s Name on Aesthetic Judgments. Empirical Studies of the Arts, 34(1), 126–139.

Crotta, A. and Vermeylen, F. (2020). Does Nudity Sell? An Econometric Analysis of the Value of Female Nudity in Modigliani Portraits. ACEI WP https://pure.eur.nl/en/publications/does-nudity-sell-an-econometric-analysis-of-the-value-of-female-n

Ekelund, Robert B., Jackson, John D., and Tollison, Robert D. (2013), Are Art Auction Estimates Biased?. Southern Economic Journal: October 2013, Vol. 80, No. 2, 454– 465.

Euwe, J. and Kim Oosterlinck (2017). Art Price Economics in the Netherlands during World War II. Journal for Art Market Studies Vol 1, No 1

Hernando, E., & Campo, S. (2017). Does the Artist’s Name Influence the Perceived Value of an Art Work? International Journal of Arts Management, 19(2), 46–58.

LeBlanc, A., Sheppard, S. (2022). Women artists: gender, ethnicity, origin and contemporary prices. J Cult Econ 46, 439–481.

Nahm, J. (2010). Price determinants and genre effects in the Korean art market: a partial linear analysis of size effect. J Cult Econ 34, 281–297.

Nead, L. (1992), The Female Nude: Art, Obscenity and Sexuality, Routledge, London & New York.

Park, J., Park, J. and Park, J. H. (2022). What Type of Title Would You Put on Your Paintings?: The Impact on the Price of Artwork According to Its Title. Empirical Studies of the Arts, Vol. 40(1) 57–80.

Radermecker, Anne-Sophie V. E. (2019). Artworks without names. Journal of Cultural Economics, Vol. 43, No. 3, pp. 443-483.

Russell, P. A., & Milne, S. (1997). Meaningfulness and Hedonic Value of Paintings: Effects of Titles. Empirical Studies of the Arts, 15(1), 61–73.

Sheppard, S. (2021). Image content, complexity, and the market value of art. Williams College Working Papers in Economics No. 2021-08.

Tan, W. R., C. S. Chan, H. E. Aguirre, and K. Tanaka, 2016, Ceci n’est pas une pipe: A deep convolutional network for fine-art paintings classification, in 2016 IEEE International Conference on Image Processing (ICIP), 3703–3707 (IEEE, Phoenix, AZ).

Vuong Q-H, Ho M-T, Nguyen H-KT, Vuong T-T, Tran K, Ho MT. “Paintings Can Be Forged, But Not Feeling”: Vietnamese Art—Market, Fraud, and Value. Arts. 2018; 7(4):62.

Reviewer #2: Overall I feel the topic of choice is interesting and the data analysis part is sufficient to support your claims.

I only have one concern about the literature review / motivation of your research question: You suggested that a limitation of the literature about painting pricing is that prior models infer themes based on titles. I am not an expert in art industry but I intuitively do not expect to infer themes from painting titles. Instead, the "visual" aspect of the painting is definitely more important. Could you please add more literature about the progress of using visual inspection, especially computer vision or machine learning approach, which is more natural in inferring painting themes and modeling art price?

6. PLOS authors have the option to publish the peer review history of their article (what does this mean?). If published, this will include your full peer review and any attached files.

Reviewer #1: No

Reviewer #2: No

---

## [Author Response · Author response to Decision Letter 0]

17 Oct 2023

We have uploaded a separate document.

Here go the comments and the responses

Dear Simon Porcher,

Thank you for inviting us to revise and resubmit our manuscript. We also want to thank you for giving us an extension as following up on one of the suggestions of reviewer #1 took us quite some time and effort. We feel it was well worth it. We have addressed all the issues raised in the editorial comments as well is those by the reviewers as best as we could. A detailed response is below. We have a version with all the changes marked (‘Revised manuscript with track changes’), as well as a cleaned version (‘Manuscript’)

We very much do hope you will find the analysis interesting and relevant.

Thank you very much,

Also on behalf of my co-authors,

AAAAAAAAAAA

Journal Requirements:

We did use the instructions provided. 

2. Thank you for stating the following financial disclosure: "No specific funding was received for this work"

The authors received no specific funding for this work 

Thanks, we provide a file with an anonymized dataset as supplementary material

4. Please include a separate caption for figure 1 in your manuscript.

We provide a caption for figure 1

5. We note that Figure 1 and Supporting Figures A1 to A14 in your submission contain copyrighted images. All PLOS content is published under the Creative Commons Attribution License (CC BY 4.0), which means that the manuscript, images, and Supporting Information files will be freely available online, and any third party is permitted to access, download, copy, distribute, and use these materials in any way, even commercially, with proper attribution. For more information, see our copyright guidelines: http://journals.plos.org/plosone/s/licenses-and-copyright.

We have uploaded these forms as “Other” 

We have included the captions

We include captions for supporting files and the end of the manuscript and updated in-text citations

 

Additional Editor Comments:

Reviewer 1 particularly highlight some major comments - e.g. on some claims, or selection effect in the sample -you should deal with. 

We address all comments, see below. 

5. Review Comments to the Author

Reviewer #1: The paper “The art of valuation: incorporating expert insights to price classical paintings by Swedish masters” focuses on an important point regarding artworks’ evaluation: the theme of the artwork. This characteristic is likely to be crucial in how collectors and other market agents evaluate artworks on sale in the market, and previous studies have investigated this issue. However, as rightly stated by the authors, it must be stressed that themes are not always easy to identify unless a visual inspection of the artwork is made. Moreover, this is not always the case, as expertise is sometimes required to accurately attribute a theme to an artwork. As a motivation for this study, I find that the authors are raising an intriguing point, contributing to the cultural economics analysis on the topic.

However, I think that the paper has some issues that should be considered. In what follows, I will outline a series of points that I believe should be considered by the authors.

Thank you very much for taking the effort to review our manuscript and to provide detailed and constructive comments that helped us improve the manuscript. In particular, we feel encouraged to make a systematic comparison between ‘visual’ and ‘semantic’ analysis of the sample artworks.

1) While, as stated above, the point raised by the authors is relevant, the results and discussion solely focus on the information presented in the table, confining the analysis to the statistical significance of the coefficients alone and to the test for the sample selection. Although there are undoubtedly important, I perceive a gap between the study's objective and the ensuing discussion. In the discussion, the authors should explicitly emphasize that: i) the theme of the artworks significantly influences their value, as established in prior literature, and ii) expert inspection should take precedence over terminological analysis based on the artwork’s title. While the first claim is already possible with the results contained in the paper, the same cannot be said for the second claim. In fact, the paper lacks a comparison of the outcomes produced by the 'terminological approach. To truly highlight the significance of expert inspection in art market evaluation, I believe the paper should incorporate this comparison, otherwise a significant disparity exists between the research question and the analysis. This comparison could also suggest the existence of an impact of measurement error on the estimated coefficients. As for now, there is no proof of an impact of this measurement error. In fact, the comparison between the two methods is only made in the inspection of the 10 paintings the authors report in the appendix, but this does not directly relate to the prices associated with specific themes in the market: this ex-ante evaluation is made outside the market and, while it supports the claim that expert inspection could be preferred to terminological deduction of the theme based on the title (although it is only based in few data points), in order to mitigate measurement error, it could not be extended to the impact on the prices.

Making the suggested comparison between the results obtained using the two methods (and clarifying that the ex-ante evaluation is not directly linked to the prices) would bring greater focus to the authors' suggested innovation, which aligns with the approach commonly used in auction houses and galleries when estimating the value of artwork. This last point should be raised as well.

Thank you very much for bringing this up. It has stimulated us to do additional research and we spend several weeks gathering the artworks’ titles. These were used to arrive at thematic classification along the method suggested and employed in Renneboog and Spaenjers (2013). It shows that a lot of works could not be classified. More specifically, it showed that twothirds could not be put into one of Garay’s themes based on their terminological analysis. We contacted Luc Renneboog and he informed us that this usually is the case with the semantic approach. For us, this was additional motivation to continue with our line of research, i.e., trying to include artistic informed characteristics in the analysis. However, it also resulted in additional work and we needed to ask for an extension, which was granted by the editors.

We can substantiate the claim of measurement error in much more detail. But most obvious is the fact that semantic classification along themes result in most paintings ending up ‘unclassified’ and that we show there is a systematic bias with semantic classification.

As a result, we feel we are now able to provide a comparison between semantic and artistic thematic classification. Please realize this has a major impact on the research design and organization of the paper.

2) The authors state (p. 12, line 266 and following) that they make use of Garay’s (2021) classification for their themes. Indeed, Garay also makes use of other themes, such as “Abstract”, “Others”, “Religion”, “Self-portrait”, and “Untitled”, which are not part of the classification used by the authors; did the authors drop the paintings falling into these 5 classes? Or it is the case that any of the art pieces analyzed fell into any of these classes? This should be clarified.

Thanks for highlighting this. We want to point out that the themes “Abstract”, “Religion”, and “Self-portrait” do not occur in the sample paintings. The categories “Others” and “Untitled” do not occur in our analysis due to the art-informed (visual) classification. Hence, we did not drop any paintings. This is now specified in the text.

Further, the authors’ point (7) called “Still life” also contain animals, which are also part of point (1): this can lead to confusion (see page 12, line 269: “still life (e.g., flowers, animals, food)”.

Thanks for raising this issue. We included a footnote to clarify.

Another possible issue related to this classification (and Garay’s as well) is that I points (3) “nude”, (5) “people”, and (6) “portrait” are all actually about people. Point (5) should then be “people” excluding “nude” and “portrait”, otherwise an artwork of a nude portrait (Anders Zorn made nude portraits and is likely to be part of the used dataset, being other of his paintings in this paper’s Figure 1) could end up in any of the three groups. The authors should explicitly state the contents of each chosen class to enhance clarity and ensure replicability of the analysis. Without this information, it is difficult to understand how the analysis was conducted and replicate the study. Additionally, it would be valuable for the authors to include the perspective of the experts who conducted the visual inspection regarding the classification employed in the paper. This would provide valuable insights and further strengthen the study.

Thanks for bringing this up. We specify in more detail and use parentheses to avoid conflusion.

3) Concerning the inspection, the title and the abstract seem to suggest that this is made by experts and art specialists. However, only a portion of the inspection was conducted by these experts. Some of the artwork themes may be easily identifiable even without advanced expertise, but I am wondering who actually made the inspection of the pieces not examined by the experts. Were the authors responsible for the inspection? If so, how was it conducted? If multiple individuals conducted the inspection for pieces not examined by the experts, did they perform it independently and compare their chosen themes for consistency? How many pieces were inspected by the experts and how many by the others? All this information should be shared.

Moreover, it is important to highlight in the abstract and introduction that not all inspections were conducted by experts. Furthermore, it is inaccurate to claim that the experts improve the classification compared to the terminological approach if some of the inspections were conducted by non-experts.

Prior studies have used the title of a painting which then enabled a classification of the art work into different themes (e.g., Renneboog and Spaenjers (2013) and Garay (2021)). Importantly, and as previously mentioned, a painting can have a title which is not related to what the art work actually shows. To overcome this obstacle in the classification, we decided to manually inspect and classify all paintings in our sample and use the same thematic groups as in Renneboog and Spaenjers (2013). Some paintings are easy to classify into a theme whereas other are more difficult. For instance, paintings from Helmer Osslund reflects in general a landscape whereas most works from Jacob Hägg shows warships; these paintings are classified into the group landscape and object, respectively. However, some paintings are more problematic to classify into a theme and in these cases, we obtained advice from art professionals from the department of Art at Uppsala university, Campus Gotland which, following inspection of the art, classified the painting into one of the eight groups used in our study. In this regard, we acknowledge the help from Maria Brunskog, Susanna Carlsten, Emily Stammers and Lars Wängdahl. A few assessments consider a painting for multiple themes (e.g., as either a landscape or an animal). These art works are included in our data following discussions with the authors and classified into one theme. For instance, a painting from Bruno Liljefors showing a fox in a winter landscape, the fox is in the centre of the painting, and hence, the painting is classified into the group animal. We amended the abstract and introduction regarding the role of experts and removed the claim that the experts improve the classification.

4) Concerning the methodology used (hedonic OLS regression and Heckman’s model), I do not think there is need to report OLS results in the paper, since the authors find that a sample selection bias exists. I think the OLS results could be removed or, at most, relegated to the appendix.

We agree with the reviewer and removed the OLS results from the tables. We do report the Heckman results only and compare the findings with paintings’ thematic classification from visual inspection with that based on semantic analysis. As said, semantic analysis results in a lot of paintings ‘unclassified’, and we investigate if there is a systematic bias. 

5) The authors state that they are following recent literature and introduce the artist’s characteristics in their analysis. This is actually not so recent (see for example Nahm 2010, Anderson et al. 2016). What is more recent is the use of artist fixed effects to capture the artist’s name effect on his/her artwork price (Cleeremans et al. 2016, Hernando and Campo 2017, Radermecker 2019, Angelini et al. 2023). I think the authors should also add this fixed effect to their analysis, since previous studies found that this effect exists, and it is also likely to influence the theme effect the authors are investigating (e.g., a nude by Zorn is likely to be sold at a higher price than a nude by Eugène Jansson).

We appreciate this suggestion. All of the analysis performed in the paper now include artist fixed effects. It is important to note that the introduction of artist fixed effects resulted in removing one of the exclusion restrictions from our Heckman model (lag of total paintings sold by artist).

6) Since this paper mostly falls and relates to cultural economics literature, the authors should expand the coverage of previous papers in this strand of literature. In fact, the analysis of themes and topics (sometimes referred to as "genre") is not new (e.g., Bocart & Oosterlinck, 2011, among many others, besides the other papers already cited by the authors), and the same is true for the use of the Heckman model, mostly after Collins et al. (2009) paper. See for example Bocart and Oosterlinck (2011), Ekelund et al. (2013), and Angelini et al. (2022). Some scholars also focused on the importance of the title (e.g., Park et al. 2022, Russell and Milne 1997). Recent works (Tan et al. 2016; Sheppard 2021; Choi et al. 2023) also implement what the authors suggest in their future extensions (or something along these lines), namely “use artificial intelligence” (p. 21, line 495).

We thank the reviewer very much for suggesting these studies. We include them in the analysis. They helped us a lot in specifying that our contribution especially is within the comparison between visual and semantic analysis.

In addition to that, some of the claims of the authors are inaccurate and might need additional support from previous studies or additional information. I list these in the following:

a. Page 8, lines 178-180: “This is because artists that have previously sold a painting are more likely to sell again at an upcoming auction.” This period needs a supporting reference or a source. 

After including artist fixed effects in the analysis, this exclusion restriction has been removed. 

b. Page 10, lines 221-222: “We exclude a small number of paintings withdrawn from auctions prior to the event when their provenance was not clear.” What a “non clear” provenance means should be explained. Is it that the published provenance in the catalogue was then deemed to be wrong? Or else? Also, how many paintings have been excluded for this reason should be stated. The same should be stated for the other reasons of exclusion (p. 9, line 211 to p. 10 line 214). 

Following advice from Professor Brunskog, we excluded 22 paintings. The reasons: a) not sure of the artist (8 paintings), b) the method used, 8 paintings, ( i.e. not examined without the frame, c) we also excluded 6 paintings with provenance unclear ( reasons not disclosed but could be theft ( stolen goods). Furthermore, we excluded four paintings with two artists, three that were painted on textiles, two painted on door-shield, as well as 819 etchings (as we wanted to keep the sample homogenous; about paintings only).

c. Page 10, line 235: “all auction prices exclude fees”. The hammer price actually excludes the buyer’s premium (which is added upon the hammer price), but the seller’s commission is usually part of the hammer price. See Ashenfelter and Graddy (2003). 

Modified

d. Page 11, line 244: “When a painting is bought by an auction house, its future price may be affected”. It is not clear what this means. 

Addressed

e. Page 15, line 358: the authors refer to the signature as a proof of authenticity. Forged artworks are an issue in the art market, and a lot of these are signed (with a forged signature). See for example Vuong et al. (2018) and Euwe and Oosterlinck (2017). A proof of authenticity would be a certificate of authenticity, not the signature (so the period at page 16, lines 360-361 should be modified). 

Modified

f. Page 4, line 87-88: “paintings by male artists sell at considerably higher prices than those from female artists”. Adams et al. (2021), already cited in the paper, should be recalled in the comparison. Other papers to use in the comparison are Cameron et al. (2019) and LeBlanc and Sheppard (2022). 

Modified

7) The authors claim that the Swedish art market can be considered a representative sample of the general art market. However, I have reservations about this assertion, as the art market is known for its significant diversity. Furhtermore, if we limit the comparison to the average selling prices, in this paper this is 215,400 SEK (around 20,000$), while the average price in other papers is way higher (see for example the average of prices in Collins et al. (2009), 223,954.800$, or the one in Renneboog and Spanjers (2013), 159,354$; both papers focus on sales in the most important international auction houses). In fact, the Swedish market is not comparable to the New York, the London, or the Paris market. I believe the authors should reconsider the extent of their claim regarding the generalizability of their analysis. While this adjustment does not diminish the research question's interest, asserting that their results apply to the entire art market is inaccurate.

The referee is correct and we deleted the claim that the Swedish art market is a representative sample of the general art market and modified the text.

Some additional minor points (which need anyway to be clarified) follows:

A) The authors find some unconventional results in their analysis, which should be discussed (and compared to previous results). Examples are:

a. watercolors fetch on average a higher price in the Swedish art market. Could it be that only very high-quality watercolors enter the market represented by the three auction houses the authors are considering, while lower-level watercolors are relegated to other auction houses or dealers? 

This no longer holds. With the introduction of artist fixed effects, and removal of one of the exclusion restriction, we now observe that watercolor paintings (similar to mixed and pastel) fetch a lower prices compared to oil paintings. This results follows what is suggested in previous literature. Namely, that oil paintings are more expensive than paintings using other media (Marinelli & Palomba, 2011), and have generally outperformed paintings using other media (Renneboog & Spaenjers, 2013).

b. As a larger fraction of paintings was sold at the preceding auction, the probability of sale declines (p. 19, lines 432-433). What could be an explanation of this? 

The text has been updated 

B) The authors claim at page 4, lines 92-93, that one of their novel contributions is that they provide detailed evidence that differences exist between sold and unsold pieces. This is actually something that is implicated by any previous results in the literature using the Heckman model and finding that a sample selection issue is at work (see point 6 above). The claim should be modified.

Modified

C) Concerning the period “Some issues with the hedonic approach include assumptions that there are no changes in the parameters across the sample period, and that it follows an implicit functional form” (p. 5, lines 111-112), it is not clear what “it” in “it follows” refers to. 

Modified

D) At p. 6, line 126, the authors state that the dependent variable of equation (1) is the price, but it is actually the log of the price. Same issue at page 10, line 232. 

Modified

E) At p. 9, lines 205-208, the authors state that online databases have a series of issues that motivated them to not choosing these, and that these databases have “problematic” classification of paintings. Both claims should be better explained, since they are quite vague. 

Addressed

F) It would be better to keep the same unit of measurement for the surface throughout the paper. Sometimes m^2 are used, some other cm^2. Cm2

Modified

G) At page 21, line 489, the authors report the claim “sex sells” as an explanation of their result related to the nude premium. However, nude in art and sex are far to be the same thing, unless one expects all nude art to be sexualized (which I think is quite unlikely). A wide discussion on this point can be found in Nead (1992). 

We took out this remark

H) In table 1, the authors sum the number of artists per year (column 3). I do not think summing these numbers make sense, since some artists have surely been exchanged in more than one year. 

Table 1 has been updated to reflect the aggregate number, rather than per year. 

References

Anderson, S.C., Ekelund, R.B., Jackson, J.D. and Tollison, R.D. (2016). Investment in early American art: the impact of transaction costs and no-sales on returns. J Cult Econ 40, 335–357. https://doi.org/10.1007/s10824-015-9252-7

Angelini, F., Castellani, M. & Pattitoni, P. (2022). You can’t export that! Export ban for modern and contemporary Italian art. Eur J Law Econ. https://doi.org/10.1007/s10657-022-09759-0

Angelini, F., Castellani, M., & Pattitoni, P. (2023). Artist Names as Human Brands: Brand Determinants, Creation and co-Creation Mechanisms. Empirical Studies of the Arts, 41(1), 80–107. https://doi.org/10.1177/02762374211072964

Ashenfelter, Orley and Kathryn Graddy. 2003. "Auctions and the Price of Art." Journal of Economic Literature, 41 (3): 763-787. 

Bocart, F. and Kim Oosterlinck (2011). Discoveries of fakes: Their impact on the art market. Economics Letters, Volume 113, Issue 2, Pages 124-126. https://doi.org/10.1016/j.econlet.2011.06.013

Cameron, L., Goetzmann, W.N. & Nozari, M. (2019). Art and gender: market bias or selection bias?. J Cult Econ 43, 279–307. https://doi.org/10.1007/s10824-019-09339-2

Choi, J., Lan Ju, Jian Li, and Zhiyong Tu (2023). Information extraction and artwork pricing. arXiv:2302.08167. https://doi.org/10.48550/arXiv.2302.08167

Cleeremans, A., Ginsburgh, V., Klein, O., & Noury, A. (2016). What’s in a Name? The Effect of an Artist’s Name on Aesthetic Judgments. Empirical Studies of the Arts, 34(1), 126–139. https://doi.org/10.1177/0276237415621197

Crotta, A. and Vermeylen, F. (2020). Does Nudity Sell? An Econometric Analysis of the Value of Female Nudity in Modigliani Portraits. ACEI WP https://pure.eur.nl/en/publications/does-nudity-sell-an-econometric-analysis-of-the-value-of-female-n

Ekelund, Robert B., Jackson, John D., and Tollison, Robert D. (2013), Are Art Auction Estimates Biased?. Southern Economic Journal: October 2013, Vol. 80, No. 2, 454– 465. https://doi.org/10.4284/0038-4038-2012.087

Euwe, J. and Kim Oosterlinck (2017). Art Price Economics in the Netherlands during World War II. Journal for Art Market Studies Vol 1, No 1, 47-67

Hernando, E., & Campo, S. (2017). Does the Artist’s Name Influence the Perceived Value of an Art Work? International Journal of Arts Management, 19(2), 46–58.

LeBlanc, A., Sheppard, S. (2022). Women artists: gender, ethnicity, origin and contemporary prices. J Cult Econ 46, 439–481. https://doi.org/10.1007/s10824-021-09431-6

Nahm, J. (2010). Price determinants and genre effects in the Korean art market: a partial linear analysis of size effect. J Cult Econ 34, 281–297. https://www.jstor.org/stable/41811061

Nead, L. (1992), The Female Nude: Art, Obscenity and Sexuality, Routledge, London & New York.

Park, J., Park, J. and Park, J. H. (2022). What Type of Title Would You Put on Your Paintings?: The Impact on the Price of Artwork According to Its Title. Empirical Studies of the Arts, Vol. 40(1) 57–80. https://doi.org/10.1177/0276237421994700

Radermecker, Anne-Sophie V. E. (2019). Artworks without names. Journal of Cultural Economics, Vol. 43, No. 3, pp. 443-483. https://doi.org/10.1007/s10824-019-09344-5

Russell, P. A., & Milne, S. (1997). Meaningfulness and Hedonic Value of Paintings: Effects of Titles. Empirical Studies of the Arts, 15(1), 61–73. https://doi.org/10.2190/EHT3-HWVM-52CB-8

Sheppard, S.C. (2021). Image content, complexity, and the market value of art. Department of Economics, Williams College. Working Papers in Economics No. 2021-08.

Tan, W. R., C. S. Chan, H. E. Aguirre, and K. Tanaka, 2016, Ceci n’est pas une pipe: A deep convolutional network for fine-art paintings classification, in 2016 IEEE International Conference on Image Processing (ICIP), 3703–3707 (IEEE, Phoenix, AZ). DOI: 10.1109/ICIP.2016.7533051

Vuong Q-H, Ho M-T, Nguyen H-K. T, Vuong T-T, Tran K, Ho M. T. “Paintings Can Be Forged, But Not Feeling”: Vietnamese Art—Market, Fraud, and Value. Arts. 2018; 7(4):62. ; https://doi.org/10.3390/arts7040062

Reviewer #2: Overall I feel the topic of choice is interesting and the data analysis part is sufficient to support your claims.

I only have one concern about the literature review / motivation of your research question: You suggested that a limitation of the literature about painting pricing is that prior models infer themes based on titles. I am not an expert in art industry but I intuitively do not expect to infer themes from painting titles. Instead, the "visual" aspect of the painting is definitely more important. Could you please add more literature about the progress of using visual inspection, especially computer vision or machine learning approach, which is more natural in inferring painting themes and modeling art price?

Thank you very much for your assessment and suggestion. 

We add more detail about the process of visual inspection and we also included additional references in relation to visual inspection of artwork (a.o., Choi et al. 2023; Sheppard, 2021; Tan et al., 2016).

---

## [Decision Letter · Decision Letter 1]

10 Nov 2023

PONE-D-23-12660R1The Art of Valuation: Using Visual Analysis to Price Classical Paintings by Swedish MastersPLOS ONE

Dear Dr. Scholtens,

Thank you for submitting your manuscript to PLOS ONE. After careful consideration, we feel that it has merit but does not fully meet PLOS ONE’s publication criteria as it currently stands. Therefore, we invite you to submit a revised version of the manuscript that addresses the points raised during the review process. Most comments are easy to address and regard some clarifications and some extra work on the format of the paper upon publication. Please address the comments, particularly those related to the methodology in order to improve the paper in its current form.

We look forward to receiving your revised manuscript.

Kind regards,

Simon Porcher

Academic Editor

PLOS ONE

Journal Requirements:

Reviewers' comments:

Reviewer's Responses to Questions

**Comments to the Author**

1. If the authors have adequately addressed your comments raised in a previous round of review and you feel that this manuscript is now acceptable for publication, you may indicate that here to bypass the “Comments to the Author” section, enter your conflict of interest statement in the “Confidential to Editor” section, and submit your "Accept" recommendation.

Reviewer #1: All comments have been addressed

Reviewer #2: (No Response)

2. Is the manuscript technically sound, and do the data support the conclusions?

Reviewer #1: Yes

Reviewer #2: Yes

3. Has the statistical analysis been performed appropriately and rigorously? 

Reviewer #1: No

Reviewer #2: Yes

4. Have the authors made all data underlying the findings in their manuscript fully available?

Reviewer #1: Yes

Reviewer #2: Yes

5. Is the manuscript presented in an intelligible fashion and written in standard English?

Reviewer #1: Yes

Reviewer #2: Yes

6. Review Comments to the Author

Reviewer #1: The authors addressed all the comments from the previous review and introduced a comparison between terminological and visual inspection of the artworks, resulting in interesting findings.

In what follows, I report a series of points that I think the authors should consider:

1) The authors introduced an artist fixed effect, which allowed them to align their results with previous studies, since a name effect is actually important in shaping the prices. However, the authors discuss the different prices obtained by male artists' pieces with respect to female artists' ones, using a dummy in their estimations. This is strange, since the artist fixed effect should capture all the artist-specific characteristics (among which gender), so every artist-specific variable could not actually be estimated if an artist fixed effect is introduced. The reason why the gender dummy has actually a coefficient notwithstanding is only reported in the footnote of the Tables 3 and 4 and the Table in appendix F, namely the authors pooled together the artists with less than 10 pieces, however this suggests that the gender dummy and the artist fixed effects are likely to be highly correlated (there is no perfect multicollinearity since some artists are pooled together, but there would be perfect multicollinearity if one would run the regression without the pooled artists). This likely inflates the standard errors, and the issue is likely to be worse as the number of pooled artists is lower. I believe that omitting the gender dummy would be the most effective means to solve this issue.

2) I appreciate the effort the authors made in carrying on the terminological analysis and the comparison with the visual inspection analysis. I think it should be made clearer, though, which was the language used. In particular, were all the artworks' title in Swedish? If so, did the authors traduce the titles in English and/or French to use the Renneboog & Spaenjers (2013) methodology (namely, the keywords they report in their Appendix B)? Or has the methodology been applied after translating the keywords for each theme in Swedish? This information is needed to allow for the replicability of the study. I think the authors may also want to replicate what Renneboog and Spanjers did in their Appendix B, reporting which words have been used for each theme.

Some minor points:

1) The authors state that the standard approach in the analysis of artwork prices has been to focus on sold paintings only (page 6, line 71 and page 7, line 100). In the previous round of review I listed some exception, as an example, but after the paper of Collins et al. (2009), a lot of articles started taking into account the existence of a possible selection bias (Google Scholar lists 87 citations of Collins et al. 2009, as for today). So it might not be appropriate to state that focusing on sold artworks only is the standard approach, since several works in the last years take selection bias into account.

2) I think uniforming the terminology throughout the paper would help the reader: sometimes the terminilogical approach is called semantic method or semantic approach; similarly, the art-informed inspection is referred to as visual inspection (or as "artistic information" in Appendix C, which I think is even misleading).

3) On page 13-14, lines 257-261, the authors refer to visual inspection themes, reporting the percentage of each theme. However, the visual inspection themes are introduced only later on (page 15). It would maybe be better to report these percentages after the introduction of the themes.

4) Appendix E contains a table which reports the matching labels using the two methods. The comparison is made using only the subsample for which the terminological approach attaches a label different from "N/A" to the artworks. I think that it might be more informative to have the table reporting the matching labels for all the sample, since the coefficients currently reported in the table may suggest that the matching is not that bad. This would also better highlight the interesting result the authors find, about measurement error (page 23, line 476-478).

5) In the paper, lambda is incorrectly spelled as "lamba" (page 23, lines 479 and 481).

6) On page 20, line 418, "tech (pastel)" should read "technique (pastel)".

7) On page 21, line 443-444, the authors write "unsigned works deliver on average a 34% discount compared to unsigned work." It should be "compared to signed works".

Reviewer #2: Though you provided three additional citations to discuss the state-of-the art techniques in extracting painting themes, I don't feel the current response is adequate in elaborating these recent progresses. I also noticed that R1 raised a similar point that "Recent works(Tan et al. 2016; Sheppard 2021; Choi et al. 2023) also implement what the authors suggest in their future extensions (or something along these lines), namely “use artificial intelligence” (p. 21, line 495)." And you simply cited R1's suggested references without clear elaboration. I would recommend spend a few paragraphs in the front part of your manuscript to introduce these progress, discuss them in detail, compare those works with your project. This will help readers better position the contributions of this work compared to traditional and most cutting edge techniques.

7. PLOS authors have the option to publish the peer review history of their article (what does this mean?). If published, this will include your full peer review and any attached files.

Reviewer #1: No

Reviewer #2: No

---

## [Author Response · Author response to Decision Letter 1]

6 Dec 2023

PONE-D-23-12660-R.1

The Art of Valuation: 

Using Visual Analysis to Price Classical Paintings by Swedish Masters

Dear Editor.

Thank you very much for your invitation to revise and resubmit our paper. We appreciate the reviewers’ comment and suggestions and did our utmost best to comply and to integrate in the revision. A detailed response goes below. 

We hope you and the reviewers find the points raised adequately addressed and we look forward to your response.

Yours sincerely,

Reviewer #1: 

The authors addressed all the comments from the previous review and introduced a comparison between terminological and visual inspection of the artworks, resulting in interesting findings. In what follows, I report a series of points that I think the authors should consider:

1) The authors introduced an artist fixed effect, which allowed them to align their results with previous studies, since a name effect is actually important in shaping the prices. However, the authors discuss the different prices obtained by male artists' pieces with respect to female artists' ones, using a dummy in their estimations. This is strange, since the artist fixed effect should capture all the artist-specific characteristics (among which gender), so every artist-specific variable could not actually be estimated if an artist fixed effect is introduced. The reason why the gender dummy has actually a coefficient notwithstanding is only reported in the footnote of the Tables 3 and 4 and the Table in appendix F, namely the authors pooled together the artists with less than 10 pieces, however this suggests that the gender dummy and the artist fixed effects are likely to be highly correlated (there is no perfect multicollinearity since some artists are pooled together, but there would be perfect multicollinearity if one would run the regression without the pooled artists). This likely inflates the standard errors, and the issue is likely to be worse as the number of pooled artists is lower. I believe that omitting the gender dummy would be the most effective means to solve this issue.

We first want to thank the reviewer for thoroughly studying our paper again and providing such thoughtful and helpful comments. As to the issue of the fixed effects in relation to artist gender, we find the reviewer is absolutely right. The inclusion of a gender dummy is of little meaning in the current setting with artist fixed effects. The dummy for gender has therefore been removed from the analysis, as the effect is rightly captured by the inclusion of artist fixed effects. Furthermore, we want to point out that pooling the artists with less than 10 paintings together is needed given the structure of the data. In our data from the three auction houses, we have a large number of artists, many of whom have sold a very limited number of paintings. Often this is only one painting, or if more than one, then they were all sold at the same auction. We therefore have a setting with many small, and unbalanced group sizes. We are therefore unable to identify any within group variation for such artist (an unidentifiability problem). Furthermore, in a setting with a limited dependent variable model estimated via maximum likelihood (such as the first stage in a Heckman selection model), including fixed effects when group sizes are small will substantially bias the estimates away from zero, with the bias diminishing with increasing group size (see also Greene, 2004). 

One solution to this issue is to estimate a random effects Heckman selection model. However, despite the potential increased efficiency of such estimator, it seems very unreasonable in this setting to assume no correlation between the artist fixed effects, and the remaining right-hand side variable. Thus, such an approach would lead to biased and inconsistent estimates. Another solution is to combine the small groups, which is the solution we opted for here. Many other authors are not faced with this issue, as their data has fewer number of artist and/or more observations per artist, with sales spread out over multiple time periods, or they only differentiate if a painting has a name/signature or not. Examples are Angelini et al (2023), Hernando & Campo (2017), Ma et al. (2022), Radermecker & Anne-Sophie (2019), Renneboog & Spaenjers (2013). 

2) I appreciate the effort the authors made in carrying on the terminological analysis and the comparison with the visual inspection analysis. I think it should be made clearer, though, which was the language used. In particular, were all the artworks' title in Swedish? If so, did the authors traduce the titles in English and/or French to use the Renneboog & Spaenjers (2013) methodology (namely, the keywords they report in their Appendix B)? Or has the methodology been applied after translating the keywords for each theme in Swedish? This information is needed to allow for the replicability of the study. I think the authors may also want to replicate what Renneboog and Spanjers did in their Appendix B, reporting which words have been used for each theme.

In our classification of paintings into topics, we use the titles as reported in the auction catalogues. With a few exceptions, all titles are in Swedish. An example of a painting in another language is when the painting represents a person, for example the painting “Mrs Emily Crane” by Anders Zorn. The classification of each painting is carried out after translating the titles and keywords from Renneboog and Spaenjers (please see their Appendix B) into Swedish. Although we follow their methodology, we use the full name of each painting. We believe this approach is appropriate as several paintings titles begins with “On the way”, “Art to” etc. We did not detect, following visual inspection, paintings which could be attributed to ABSTRACT, RELIGION, or UNTITLED (due to the sample paintings). However, we added a new category, OBJECT, as it illustrates a “wall”, “sailboat”, “boat”, “a lighthouse”, “castle”, “beehive”, “apple”, “orange”, “grape”, “rain” or “cloud”. The classification into topic categories was executed using Swedish.

The categories and the search strings used in our paper is as follows:

ANIMAL (horse, bird, elk, fox, dog, cats, sheep, hen, cock, geese, goose)

LANDSCAPE (landscape, country landscape, coastal landscape, seascape, winter landscape, snow landscape, mountain, lake, sea, cliffs, valley)

NUDE (model, nude)

OBJECT (wall, sailboat, boat, lighthouse, castle, wall, beehive, apple, orange, grape, rain, cloud)

PEOPLE (people, person, mr, mrs, herr, fru, family, boy, girl, man, woman, child, infant, kid, baby)

PORTRAIT (portrait, self-portrait)

STILL LIFE (still life, nature, flower, flower wase, bouquet)

URBAN (city, town, village, street, harbor, port, Stockholm, Visby, Öland, Arild, Bretagne, London, Paris, Västkusten) 

Some minor points:

1) The authors state that the standard approach in the analysis of artwork prices has been to focus on sold paintings only (page 6, line 71 and page 7, line 100). In the previous round of review I listed some exception, as an example, but after the paper of Collins et al. (2009), a lot of articles started taking into account the existence of a possible selection bias (Google Scholar lists 87 citations of Collins et al. 2009, as for today). So it might not be appropriate to state that focusing on sold artworks only is the standard approach, since several works in the last years take selection bias into account.

We have reformulated our statements. 

2) I think uniforming the terminology throughout the paper would help the reader: sometimes the terminilogical approach is called semantic method or semantic approach; similarly, the art-informed inspection is referred to as visual inspection (or as "artistic information" in Appendix C, which I think is even misleading).

The naming convention has been cleaned up and made consistent throughout the paper; we label as terminological approach throughout and refrain from suggesting that the method relies on artistic information.

3) On page 13-14, lines 257-261, the authors refer to visual inspection themes, reporting the percentage of each theme. However, the visual inspection themes are introduced only later on (page 15). It would maybe be better to report these percentages after the introduction of the themes.

We have moved this reference to Appendix D, so it is now better aligned with the rest of the paper on we expect it will not lead to any confusion. 

4) Appendix E contains a table which reports the matching labels using the two methods. The comparison is made using only the subsample for which the terminological approach attaches a label different from "N/A" to the artworks. I think that it might be more informative to have the table reporting the matching labels for all the sample, since the coefficients currently reported in the table may suggest that the matching is not that bad. This would also better highlight the interesting result the authors find, about measurement error (page 23, line 476-478).

Appendix E has been updated to reflect this suggestion. 

5) In the paper, lambda is incorrectly spelled as "lamba" (page 23, lines 479 and 481).

Fixed this

6) On page 20, line 418, "tech (pastel)" should read "technique (pastel)".

Fixed this

7) On page 21, line 443-444, the authors write "unsigned works deliver on average a 34% discount compared to unsigned work." It should be "compared to signed works".

Fixed this

 

Reviewer #2: 

Though you provided three additional citations to discuss the state-of-the art techniques in extracting painting themes, I don't feel the current response is adequate in elaborating these recent progresses. I also noticed that R1 raised a similar point that "Recent works (Tan et al. 2016; Sheppard 2021; Choi et al. 2023) also implement what the authors suggest in their future extensions (or something along these lines), namely “use artificial intelligence” (p. 21, line 495)." And you simply cited R1's suggested references without clear elaboration. I would recommend spend a few paragraphs in the front part of your manuscript to introduce these progress, discuss them in detail, compare those works with your project. This will help readers better position the contributions of this work compared to traditional and most cutting edge techniques.

Thank you for your suggestion. We elaborate on these developments in the revised paper’s introduction. However, we are sure you are well aware of the fact that we have to keep the study short and don’t want to suggest we envision this study to be a complete review of the literature as ours in a research paper. We refrain from referring to using AI for visual inspection. It is hard to include a comparison between AI approaches and our visual inspection as we are not aware of these techniques being applied to our sample paintings. We believe our relative advantage is in machine learning regarding artwork properties and have started to work on this already. 

 

References:

Angelini, F., Castellani, M., & Pattitoni, P. (2023). Artist Names as Human Brands: Brand Determinants, Creation and co-Creation Mechanisms. Empirical Studies of the Arts, 41(1): 80–107. https://doi.org/10.1177/02762374211072964

Greene, W. (2004). The behaviour of the maximum likelihood estimator of limited dependent variable models in the presence of fixed effects. The Econometrics Journal, 7(1): 98-119.

Hernando, E., & Campo, S. (2017). Does the Artist’s Name Influence the Perceived Value of an Art Work? International Journal of Arts Management, 19(2): 46–58.

Ma, M.X., Noussair, C.N., Renneboog, L. (2022). Colors, Emotions, and the Auction Value of Paintings, European Economic Review, 142, https://doi.org/10.1016/j.euroecorev.2021.104004.

Radermecker, A.V.E. (2019). Artworks without names. Journal of Cultural Economics, 43 (3): 443-483. https://doi.org/10.1007/s10824-019-09344-5

Renneboog, L. and Spaenjers, C. (2013) ‘Buying Beauty: On Prices and Returns in the Art Market’, Management Science, 59:36–53.

---

## [Decision Letter · Decision Letter 2]

21 Dec 2023

The Art of Valuation: Using Visual Analysis to Price Classical Paintings by Swedish Masters

PONE-D-23-12660R2

Dear Dr. Scholtens,

We’re pleased to inform you that your manuscript has been judged scientifically suitable for publication and will be formally accepted for publication once it meets all outstanding technical requirements.

Kind regards,

Simon Porcher

Academic Editor

PLOS ONE

Additional Editor Comments (optional):

Reviewers' comments:

Reviewer's Responses to Questions

**Comments to the Author**

1. If the authors have adequately addressed your comments raised in a previous round of review and you feel that this manuscript is now acceptable for publication, you may indicate that here to bypass the “Comments to the Author” section, enter your conflict of interest statement in the “Confidential to Editor” section, and submit your "Accept" recommendation.

Reviewer #1: All comments have been addressed

Reviewer #2: All comments have been addressed

2. Is the manuscript technically sound, and do the data support the conclusions?

Reviewer #1: Yes

Reviewer #2: Yes

3. Has the statistical analysis been performed appropriately and rigorously? 

Reviewer #1: Yes

Reviewer #2: Yes

4. Have the authors made all data underlying the findings in their manuscript fully available?

Reviewer #1: Yes

Reviewer #2: Yes

5. Is the manuscript presented in an intelligible fashion and written in standard English?

Reviewer #1: Yes

Reviewer #2: Yes

6. Review Comments to the Author

Reviewer #1: The authors took into account all the comments and suggestions of my report in a satisfactory manner.

Reviewer #2: There is one minor punctuation mistake when you mention "Choi et al.., 2023; ". Please fix it. I have no more comments.

7. PLOS authors have the option to publish the peer review history of their article (what does this mean?). If published, this will include your full peer review and any attached files.

Reviewer #1: No

Reviewer #2: No

---

## [Editor Report · Acceptance letter]

8 Jan 2024

PONE-D-23-12660R2 

PLOS ONE

Dear Dr. Scholtens, 

I'm pleased to inform you that your manuscript has been deemed suitable for publication in PLOS ONE. Congratulations! Your manuscript is now being handed over to our production team.

Kind regards, 

on behalf of

Pr. Simon Porcher 

Academic Editor

PLOS ONE